



**Machine-learning ensembled CMIP6 projection reveals socio-economic pathways**
**will aggravate global warming and precipitation extreme**
Piaoyin Zhang, Jianzhong Lu, Xiaoling Chen
State Key Laboratory of Information Engineering in Surveying, Mapping and Remote
Sensing, Wuhan University, Wuhan, 430072, China
*Correspondence to*: Jianzhong Lu (lujzhong@whu.edu.cn)



**Abstract:** The climate change plays a key role in ecosystem evolution and has been
proved to be affected by comprehensive factors including anthropogenic activities. The
application of GCMs (General Circulation Models) launched by CMIP6 (Coupled
Model Intercomparison Project Phase 6) has become a primary implement to catch
future climate characteristics under different future socio-economic pathways.
However, quantitative future climate change records with high credibility generated by
robust GCMs merged datasets from CMIP6 are scarce. Most precious studies depended
on traditional GCMs ensemble datasets (e.g., single, mean and medium) which were
proved to be highly unstable. In this study, three machine learning methods (Ordinary
Least Squares regression, Decision Tree, and Deep Neural Networks) were applied to
ensemble temperature and precipitation from 16 CMIP6 GCMs simultaneously.
Monthly optimal estimation of precipitation and temperature from three datasets were
selected to generate a new ensemble dataset under three Socio-Economic Pathways
(SSP1-2.6, SSP2-4.5 and SSP5-8.5). The new ensemble precipitation (temperature)
dataset with the R=0.81 (0.99) is more accurate than all the single GCM. High credible
analyses demonstrate that Europe and North America contribute more to global
warming than Oceania, Africa and South America. The global continent break through
1.5 ℃, 2 ℃ and 3 ℃ rising threshold in 2024, 2031 and 2048 under SSP5-8.5 scenarios.
Most precipitation aggregates in July and August, while dry months fall in April and
September to next February during the rest of $21^{st}$ century. Global precipitation will be
accelerated polarization with the decreasing trends of Africa and Asia ($p < 0.05$) under
the scenario of SSP5-8.5. The proposed analysis provides credible opportunities and



quantitative fundamental to understand future climate characteristics for ecology and
meteorology.

## 1. Introduction

As essential components of global climate transformation, the pattern changes of
temperature and precipitation broadly impact agricultural productivity (Iwamura et al.,
2020; Ortiz-Bobea et al., 2021; Raupach et al., 2021), ocean acidification (Randall and
van Woesik, 2015; Anthony, 2016), hydrological drought or flooding extremes (Zhang
et al., 2019; Liu et al., 2021; Qi et al., 2021) and spreading viruses (Iwamura et al.,
2020; Li et al., 2018), etc. The Paris Agreement was set for reinforcing global response
to control warming level below 2 °C and pursuing for 1.5°C impact (Hulme, 2016;
Schleussner et al., 2016) compared with the pre-industrial period (1850-1900).
However, IPCC Sixth Assessment Report (AR6) statement has affirmed that emissions
of greenhouse gases from anthropogenic activities are responsible for 1.1°C
temperature rising if 1850-1900 is defined as the baseline period (IPCC, 2021). Hence,
it is fundamental to predict climate characteristics depending on the robust projection
data set for formulating future climate change policies.

The utilization of meteorological station data or satellite products is failed to project
climate changes (Dar and Dar, 2021). However, the Coupled Model Intercomparison
Project (CMIP) has provided a great number of GCMs (General Circulation Models)
for researchers to catch future climate changes. In past decades, former CMIPs played





an active role in regional studies which were related to climate change projection. Lee
et al. (2020) indicated the rising of maximum precipitation in East Asia will exceed to
7, 15 and 35 percent under RCP2.6, RCP4.5 and RCP8.5 conditions at the end of the
21$^{st}$ century. Gaitán et al. (2019) employed 9 GCMs and demonstrated the greatest
rising daily maximum temperature over Spain will reach to 7℃ until 2100 for RCP8.5.
In the 6$^{th}$ phase of CMIP, five Socio-Economic Pathways (SSPs) which launched to
describe human development challenges (Iqbal et al., 2021; You et al., 2021; Xu et al.,
2022; O'Neill et al., 2017). The resolution and dynamic parameterization scheme of
models were also improved from CMIP5 to CMIP6 (Chen et al., 2021; Hamed et al.,
2022). However, the findings generated by new ensemble climate global dataset are
rarely reported under CMIP6 with the new emission strategy. Therefore, it is
worthwhile to further utilize CMIP6 GCMs.

Due to physical parameters sensitivity of GCMs, model outputs perform unequally
credible in a specific region or time. Climate change projection ignoring the temporal
and spatial heterogeneity leads to the incredibility of the estimation.    Utilizing only
one model will improve the uncertainty of climate projection. Therefore, ensemble
methods were widely used by taking advantage of multi GCMs. Currently, the
application of ensemble models can be roughly divided into three categories: (1) use of
individual models, average or medium combination and other traditional statistical
methods with equivalent weights (Fu et al., 2020; Li et al., 2020; Narsey et al., 2020;
Xin et al., 2020; Almazroui et al., 2021; Hermans et al., 2021), (2) new weighted





procedures with spatiotemporal homogeneity, such as independence weighted mean
(IWM) and multidimensional scaling (MDS) (Sanderson et al., 2015; Bai et al., 2021),
(3) development of machine learning (ML) with nonlinear function to train selected
models adjusted by bias correction (Xu et al., 2020; Wei et al., 2021).
Nowadays, ML applications in data-driven geoscience mainly focus on
downscaling (Tran Anh et al., 2019; Vandal et al., 2019), land cover transmission
(Condro et al., 2019; Gianinetto et al., 2020) and inversion model construction (Jiang
et al., 2019a; Liu and Grana, 2019), etc. To correct climate models, ML has been proved
to be an effective tool in taking advantage of excellent features from GCMs in several
studies (Wei et al., 2021; Jose et al., 2022). Jose. et al. (2021) employed support vector
machine in maximum temperature ensemble of CMIP GCMs with a slight improvement
of R from 0.522 to 0.7. Kuma. et al. (2022) developed an ANN network to correct cloud
feedback for CMIP5 dataset, which is superior to the mean ensemble approach, but
ANN could only explain 47% variance. Though ML methods was successfully applied
in the precious regional studies, regionalized models were just suitable for specified
periods or regions (Singh et al., 2017). Mitra (2021) anticipated there were significant
room for improvement of ML application in projection of climate variables with spatial-
temporal heterogeneity consideration. The robust application of ML application in
global climate projection based on CMIP6 GCMs is still limited and needs to be
explored.




The study aims to investigate global future climate changes based on ensemble
optimized climate datasets through ML. Firstly, the machine learning methods Ordinary
Least Square (OLS), Decision Tree (DT), and Deep Neural Networks (DNN) were used
to simulate historical global temperature and precipitation based on 16 individual
GCMs. Then, the best monthly ensemble model would be selected to project
temperature and precipitation (2015-2100) under SSP1-2.6, SSP2-4.5 and SSP5-8.5
scenarios. Finally, the tendency of global warming under 1.5℃, 2℃ and 3℃ was
explored. The precipitation pattern on a global and continental scale also be identified
under future scenarios. This study can provide scientific dataset support for scholars in
related earth science research and offer predictable opinions on climate management
measures for policy-makers.

**2. Data and Methodology**
*2.1 Experimental data*
2.1.1 Model outputs
In our study, monthly mean temperature and precipitation datasets were provided by
CMIP6 GCMs output. Sixteen GCMs developed by 19 global institutions were selected
as Table 1. The period of 1965-2014 and 2015-2100 were chosen for historical
simulation and future SSPs-RCPs scenarios, respectively. Future climate change was
projected under scenarios SSP1-2.6, SSP2-4.5 and SSP5-8.5 corresponding to the



sustainable development pathway, central pathway following the historical pattern and
fossil-intensive emission pathway (O'Neill et al., 2016), respectively. There are
different grid sizes for the selected GCMs, therefore bilinear interpolation was applied
to unify the resolution to 0.5°×0.5°.

122           Table 1 Detailed description of selected CMIP6 models

| Model Name | Modeling group | Original |
|---|---|---|
| BCC-CSM2-MR | Beijing Climate Center, China / Meteorological Administration, China | 1.125°×1.125° |
| CanESM5 | Canadian Centre for Climate Modelling and Analysis, Canada | 2.8125°×2.8125° |
| CESM2-WACCM | National Center for Atmospheric Research, Climate and Global Dynamics Laboratory, USA | 1.25°×0.9375° |
| CMCC-CM2-SR5 | Fondazione Centro Euro-Mediterraneo sui Cambiamenti Climatici Italy | 1.25°×0.9375° |
| CMCC-ESM2 | Fondazione Centro Euro-Mediterraneo sui Cambiamenti Climatici,Italy | 1.25°×0.9375° |
| FGOALS-f3-L | Chinese Academy of Sciences, China | 1.25°×1° |
| INM-CM4-8 | Institute for Numerical Mathematics, Russia | 2°×1.5° |
| INM-CM5-0 | Institute for Numerical Mathematics, Russia | 2°×1.5° |
| KACE-1-0-G | National Institute of Meteorological Sciences/Korea Meteorological Administration, Republic of Korea | 1.875°×1.25° |
| MIROC6 | The University of Tokyo, National Institute for Environmental Studies, and Japan Agency for Marine–Earth Science, Japan | 1.4063°×1.4063° |
| MRI-ESM2-0 | Meteorological Research Institute, Japan | 1.125°×1.135° |
| NESM3 | Nanjing University of Information Science and Technology, China | 1.875°×1.875° |
| TaiESM1 | Research Center for Environmental Changes, Taiwan | 1.25°×0.9375° |
| MPI-ESM1-2-HR | Max Planck Institute for Meteorology, Germany | 0.9375°×0.9375° |
| MPI-ESM1-2-LR | Max Planck Institute for Meteorology, Germany | 0.9375°×0.9375° |



| FIO-ESM-2-0 | FIO (First Institute of Oceanography, State Oceanic Administration, China), QNLM (Qingdao National Laboratory for Marine Science and Technology, China) | 1.25°×0.9375° |


2.1.2 Observation datasets
High resolution (0.5° × 0.5°) CRU TS4.05 grids (Das et al., 2016) were obtained as
monthly observation dataset for mean temperature and precipitation. Compared with
previous CRU TS4.0, the latest version CRU TS4.05 covered more complete time series
(Jan. 1901- Dec. 2020) was provided by the University of East Anglia in July
2021(Ullah et al., 2020). Considering the time-series matching problem and premature
period lack of reliability, data during period (Jan.1965- Dec.2014) were used to simulate
and validate multi-model ensemble results.

*2.2 Multi-model ensemble methods*
In the processing of multi-model ensemble, CRU TS4.05 and 16 GCMs was chosen as
ground truth and simulation dataset. Period encompassing 1965-2014 was spilt into
training period (1965-1994) and testing period (1995-2014). The input datasets are
5760 GCMs images and 540 observation images, and each image consists of 67420
pixels (Fig.1). In the training process of ensemble methods, OLS (Ordinary Least
Squares regression), DT (Decision Tree) and DNN (Deep Neural Networks) were
applied to optimize the monthly dataset.



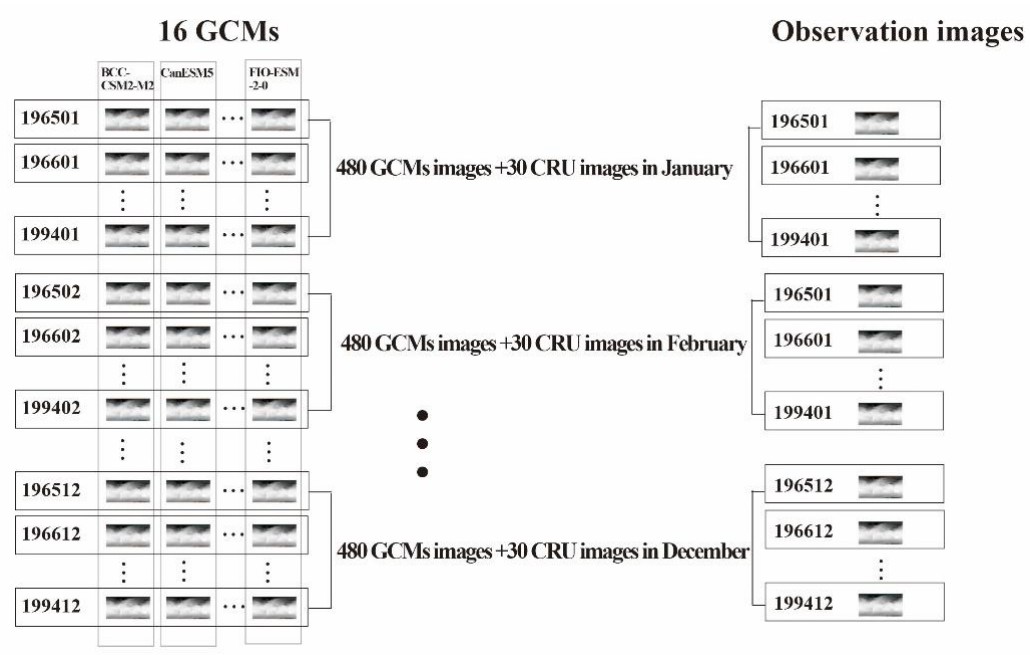

Fig. 1. Weight assignment of 16 GCMs on a time scale
The Ordinary Least Squares regression (OLS) is a widely technique applied for
estimating the unknown coefficients of linear regression equations which determine the
relationship between one or more independent quantitative variables and another
variable (Lee et al., 2022). To construct the optimization function, OLS aims to
minimize the sum of squared residuals between observed and predicted data (Sharif et
al., 2017). The OLS method was employed to assign weights for 16 selected GCMs
with linear regression at the monthly scale. The weight matrix generated by OLS can
be expressed as follow.

$$\begin{bmatrix} W^1 \\ W^2 \\ \vdots \\ W^i \\ \vdots \\ W^{12} \end{bmatrix} = \begin{bmatrix} \beta_1^1, & \beta_2^1, & \cdots, & \beta_j^1, & \cdots, & \beta_{16}^1, & \varepsilon_1 \\ \beta_1^2, & \beta_2^2, & \cdots, & \beta_j^2, & \cdots, & \beta_{16}^2, & \varepsilon_2 \\ & & \vdots & \vdots & \vdots & \vdots & \vdots & \vdots \\ \beta_1^i, & \beta_2^i, & \cdots, & \beta_j^i, & \cdots, & \beta_{16}^i, & \varepsilon_i \\ & & \vdots & \vdots & \vdots & \vdots & \vdots & \vdots \\ \beta_1^{12}, & \beta_2^{12}, & \cdots, & \beta_j^{12}, & \cdots, & \beta_{16}^{12}, & \varepsilon_{12} \end{bmatrix}$$    (1)



where $\beta_j^i$ represents the weight of the $j^{th}$ GCM in the $i^{th}$ month; $\varepsilon_i$ represents the
residual generated after weight distribution for $i^{th}$ month.

To obtain ensemble value of each pixel, the linear model generated by OLS can be
described as follow.

$$Y^{(i,k)} = \sum_{p}^{i=1} \beta_j^i X_j^{(i,k)} + \varepsilon_i \tag{2}$$

where $Y^{(i,k)}$ and $X_j^{(i,k)}$ denote the values of single $k^{th}$ pixel value in the ensemble
image and the image of $j^{th}$ GCM, respectively.

The DT method is usually applied to construct a nonlinear model which is sensitive to
intermediate missing values with stronger explanatory than linear regression (Pekel,
2020). According to the training input dataset, each region is recursively divided into
two subregions originally, in which the output value is determined to construct a binary
decision tree (Jumin et al., 2021). The process can be described as four steps in details:

Step 1: Each GCM represents a dimension of a space. Dividing the $j^{th}$ dimension of the
space into two regions (R1 and R2) by selected candidate splitting the $j^{th}$ GCM as the
feature, and then splitting the pixel values into two groups as following equations.

$$R1(j, s) = \{x \mid x(j) \leq s\} \tag{3}$$


$$R2(j, s) = \{x \mid x(j) > s\} \tag{4}$$

Step 2: Adjusting the j and s to minimize the residual sum of squares following equation
173    4.





$$min_{j,s} \left[ min_{c_1} \sum_{x_i \in R_1(j,s)} (y_i - c_1)^2 + min_{c_2} \sum_{x_i \in R_2(j,s)} (y_i - c_2)^2 \right] \qquad (5)$$

$$c_m = \frac{1}{N_m} \sum_{y_i \in R_1(j,s)} y_i \ (x \in R_m, m = 1,2) \qquad (6)$$

where $N_m$ is the total number (30 images × 67420 pixel/images) of observation data at
current node; $y_i$ is the $i^{th}$ individual sample of observation data.

Step 3: Repeating steps 1 and 2 to continue increasing the depth of tree and splitting the
subregions R1 and R2 until training loss reaches to criteria threshold. Mean-absolute-
error was applied as supported criteria to measure the quality of a split in this study.

The Deep Neural Network (DNN) is a feedforward artificial neural network, which is
applied to explore the relationship between input features and construct linear equations
for ground truth. It is an effective strategy to solve supervision problems (classification,
regression, clustering, etc.) (Raheli et al., 2017; Jiang et al., 2019b). In this study, DNN
can be split into three parts: 1input layer, 3 hidden layers and 1 output layer, meanwhile
the output of each hidden layer is transformed by the ReLU activation function. To
obtain the optimal weight of selected 16 GCMs on time scale, DNN is needed to
construct for each month. In the process of training, the method adjusts the parameters,
or the weights and biases of the model to minimize error. Our DNN neural network was
designed (Fig. 2) with 0.001 learning rate. Input $Node_i$ represents the pixel values in
the images of $i^{th}$ GCM in the form of vector [pixel$_1$, pixel$_2$, …, pixel$_m$]. Output Node
represents the pixels in the images of ensemble images in the form of vector [pixel$_1$,





pixel$_2$, ..., pixel$_m$]. Supposing there are $m$ and $n$ neurons in the $k^{th}$ and $(k+1)^{th}$ layers,
respectively, the output weight a$^k$ of the k$^{th}$ layer can be described as follow.
$$a^k = W^k a^{k-1} + b^k \tag{7}$$

where $b^k$ represents $1 \times n$ residual vector; $W^k$ represents a $n \times m$ weight matrix
composed of linear coefficient of the $k^{th}$ layer.

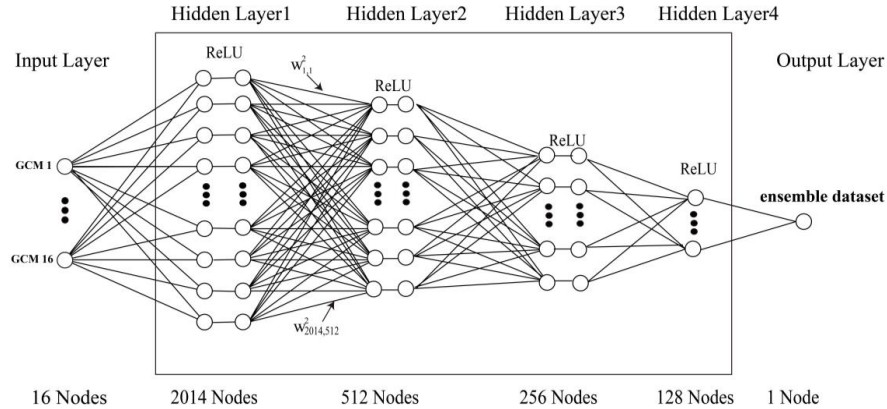

Fig. 2. Main Deep Neural Networks structure constructed in study. $\omega^l_{j,k}$ represents
the weight from the j$^{th}$ neuron in the (l-1)$^{th}$ layer to the k$^{th}$ neuron in the l$^{th}$ layer.

*2.3 Model performance assessment*
The statistic indices including correlation coefficient (*R*), centralized root mean square
difference (CRMSE), standard deviation (SD) ratio and mean absolute error (MAE) are
employed to quantify the loss between simulation and observation data. The
comprehensive rating index was applied to assess the overall result performance.

Correlation coefficient (*R*) ranging from -1 to 1 is employed to determine the linear



relationship between variables. According to $R$, correlation strength can be divided into
five degrees (Asuero et al., 2006), representing very strong ($0.7 < |R| \leqslant 1$), strong ($0.5$
$< |R| \leqslant 0.7$), moderate ($0.3 < |R| \leqslant 0.5$), weak ($0 < |R| \leqslant 0.3$) and none ($|R| = 0$)
relationships, respectively. Positive $R$ denotes variables moving in same direction and
negative $R$ represents variables move in opposite direction. The most widely applied
coefficient was generated by the Pearson product-moment correlation. $R$ is calculated
as follows (Maimon et al., 1986):

$$R = \frac{\sum_{i=1}^{n}(x_i - m_x)\left(y_i - m_y\right)}{\sqrt{[\sum_{i=1}^{n}(x_i - m_x)^2][\sum_{i=1}^{n}\left(y_i - m_y\right)^2]}} \tag{8}$$

where $x_i, y_i$ are the values of $x$ and $y$ for the $i^{th}$ individual; $m_x$, $m_y$ denote mean value of
compared variables x and y, respectively; n denotes pairs of observation and model data
matched by time-interspace.

The CRMSE and SD ratio are constructed as following equations (Taylor, 2001):

$$CRMSE = \sqrt{\frac{1}{n}\sum_{i=0}^{n}\left[(x_i - m_x) - \left(y_i - m_y\right)\right]^2} \tag{9}$$

$$SD\ ratio = \frac{\sqrt{\sum_{i=1}^{n}(x_i - m_x)^2}}{\sqrt{\sum_{i=1}^{n}(y_i - m_y)^2}} \tag{10}$$

All parameters in Equation 3 and 4 have the same meaning as Equation 2.

To evaluate the accuracy of the given model, mean absolute error (MAE) was proposed
with range of 0 to $+\infty$. The lower the value of MAE, the better a model fits the dataset,
where 0 suggests perfect simulation capability. MAE can be expressed as follows:



$$MAE = \frac{1}{N} \sum_{i=1}^{N} |y_o - y_p| \qquad (11)$$

where $y_o$ and $y_p$ represent the individual of original and predicted values, respectively;
$N$ denotes the number of observed individuals.

The assessment results of best single models or ensemble methods using different
evaluation indicators will be different. Therefore, Comprehensive Rating Index (CRI)
restricted in 0 to 1 is devised to unify standards to normalize simulation capabilities and
give concise overall ranking summary of 16 studied single models and 3 ensemble
methods (Jiang et al., 2015). The performance with CRI close to 1 is proved to be
suitable. CRI can be computed by the following formula:
$$CRI = 1 - \frac{1}{i\,j} \sum_{p=0}^{i} rank_p \qquad (12)$$

where $i$ and $j$ denote the number of evaluation indices and investigated models or
methods, respectively; $rank_p$ denotes the rank of model or method according to $p^{th}$ index.
**3. Results**
*3.1 Accuracy validation of proposed dataset by observation data in historic period*
3.1.1 Accuracy assessment of monthly averaged precipitation and temperature with
Taylor diagram
To illustrate the accuracy of 16 GCMs and 3 ensemble methods, Taylor diagram was
applied to integrate R, SD ratio and CRMSE measurements (Fig. 2). The best optimal



performance is equipped with the lowest CRMSE, highest R and SD ratio closing to 1
in Taylor diagram. Obviously, the accuracy of OLS and DNN results was better under
historical scenarios than precipitation or temperature from each GCM (Fig. 3a). Despite
slightly more excellent performance in temperature, DT method was far superior to
other single models with a significantly higher $R$ of 0.71 against CRU TS4.05
precipitation under validation period (1995-2014). The SD ratio of 16 models and 3
methods were all closed to 1 while R exceeded to 0.95. The DNN method owned the
perfect simulation with the highest $R$ of 0.985 and lowest CRMSE of 0.171 mm/month,
followed by the OLS method (R=0.983, CRMSE=0.181 mm/month) and the DT
method (R=0.972, CRMSE=0.232 mm/month). The R and CRMSE of single model
ranged from 0.956-0.971 and 0.247-0.298 mm/month. Compared with the CanESM5
model ranked as the poorest model, the DNN method reduces CRMSE by 42.7%. In
terms of precipitation (Fig. 3b), R of the OLS, DT and DNN methods were 0.800, 0.718
and 0.819, larger than other single models with a range of 0.541-0.654, respectively. R
indicated that the simulation result produced by ensemble methods owned higher
credibility. The results accuracy ranked in top three with CRMSE were still datasets
from ensemble methods DNN (CRMSE = 0.601) > OLS (CRMSE = 0.619)> DT
(CRMSE = 0.827).



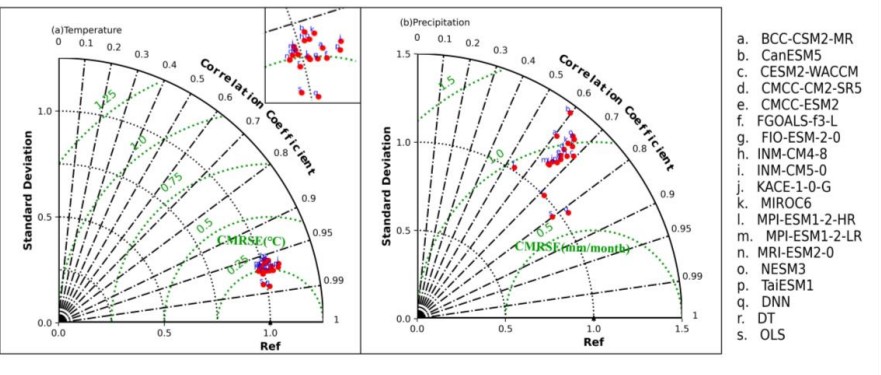


Fig. 3. Taylor diagrams of (a) temperature and (b) precipitation. Ref stands for CRU
TS4.05 observation dataset


3.1.2 Accuracy assessment by spatial pattern of MAE
To further verify the simulation performance of the single models and ensemble
methods, MAE was employed as another evaluation criterion. The value of MAE closer
to 0 indicated more precise simulation. The quantitative results were shown in Fig.4
where red lines denoted median MAE and blue lines represented mean MAE. In terms
of temperature and precipitation, the ranks of performance determined by mean MAE
were both DNN > OLS > DT > any selected single model. Moreover, median MAE of
the DNN and OLS method were 18.3 mm/month and 18.7 mm/month (1.88 ℃ and
1.96 ℃) in projecting precipitation (temperature), which showed significant robustness
of both methods.

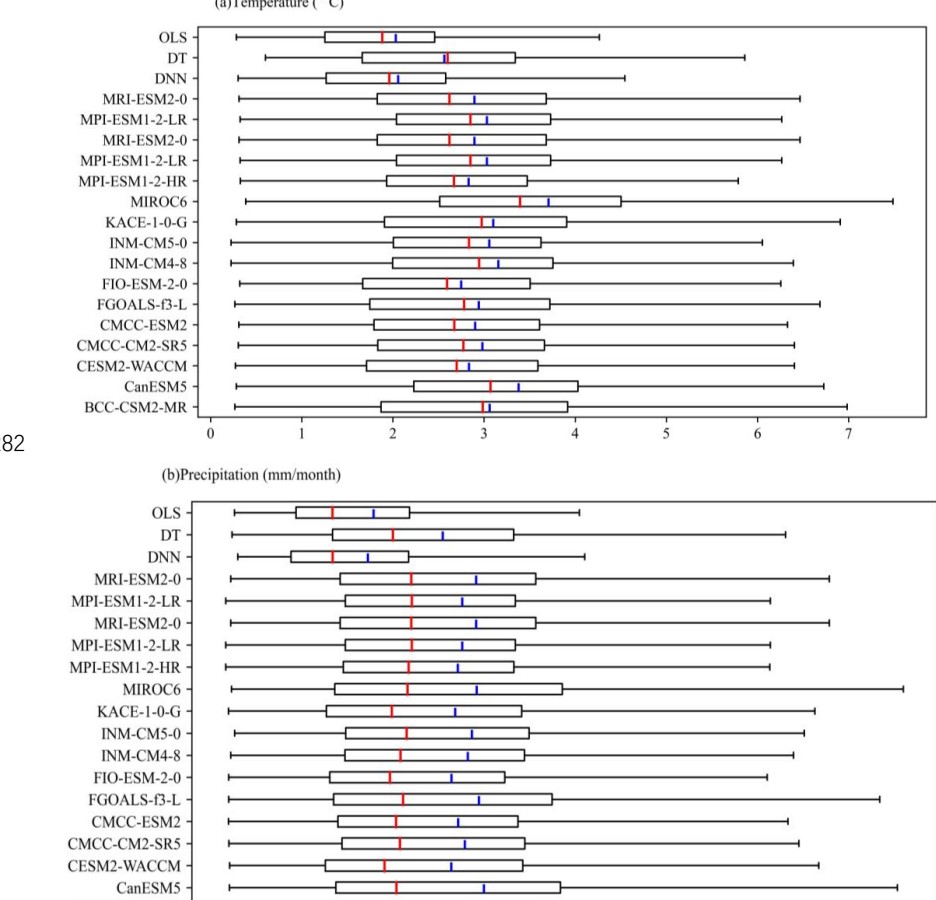



Fig. 4. Boxplots of Quantitative MAE assessment between simulation and observation
dataset for (a) temperature (℃) and (b) precipitation (mm/month). The statistical
distribution of data was displayed based on a five-divided category (minimum, first
quartile, median, third percentile and maximum).


As for temperature, MAE corresponding to each pixel (0.5°×0.5°) was mapped in Fig
5. According to the simulative mechanism, figures can be divided into two groups:
Fig5(a)-(p) and Fig5(q)-(s). The former revealed MAEs produced by 16 single models,
the latter suggested MAEs processed by ensemble methods. For 16 GCMs, with the



increase of latitude in the northern hemisphere, the area ratio with red gradually
increased, which implies the upper regions of the northern hemisphere owned higher
density of MAE. Estimation in the southern hemisphere is far better than the northern
hemisphere. Evidently, the projection each single model was far inferior to ensemble
methods. Compared with a single model, the OLS, DT and DNN methods reduced
MAE in the northern hemisphere. For example, it is obvious that the tendency of MAE
from 16 GCMs to ensemble methods decreased in Siberian plain, which locates in the
middle and high latitudes with significant continental climate. The extremely low
temperature in Siberian plain is only second to Antarctic continent, which leads to the
increasing challenge of climate change projection. There were still minor defects in the
sub-regions of the Andes Mountains in South America. The quality of the dataset
generated by different ensemble methods largely depends on the input GCMs, which is
the reason for the shortcomings in above mentioned area.

A similar MAE assessment is also conducted to precipitation. Contrary to temperature,
MAE performance of precipitation was more excellent in the northern hemisphere than
in the southern hemisphere (Fig. 6). In addition, the error showed an upward tendency
with latitude increasing in the south hemisphere. It is undeniable that ensemble methods
significantly mitigated the gap between observation and simulated gridded data
especially in southeastern Asia continent (Indian Peninsula, the Tibetan Plateau,
Thailand, etc.). Forecasts near the Andes Mountains were still unsatisfactory in
precipitation. Lack of accuracy in single model greatly amplified the difficulty of
climate change projection.

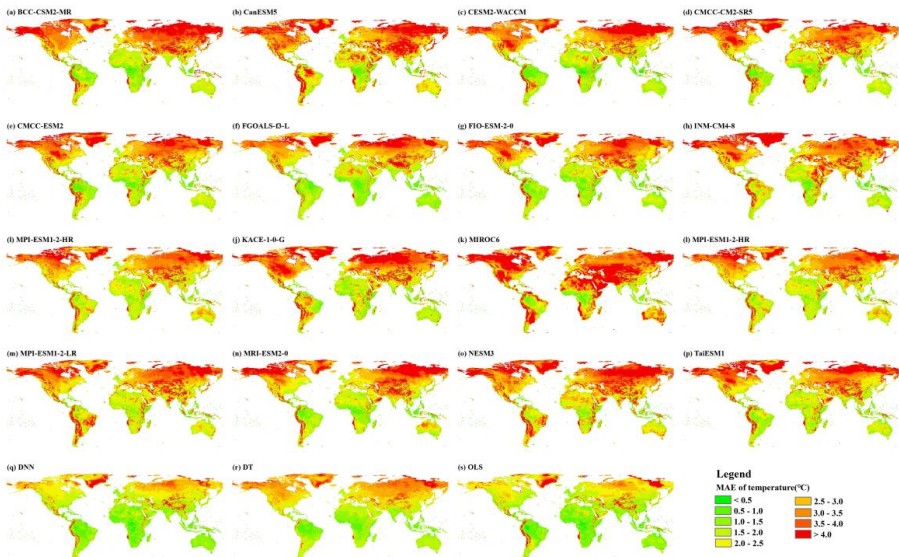

Fig. 5. The spatial distribution illustration of temperature MAE produced by selected CMIP6 models,
DNN (Deep Neural Networks), DT (Decision Tree), and OLS (Ordinary Least Squares regression).

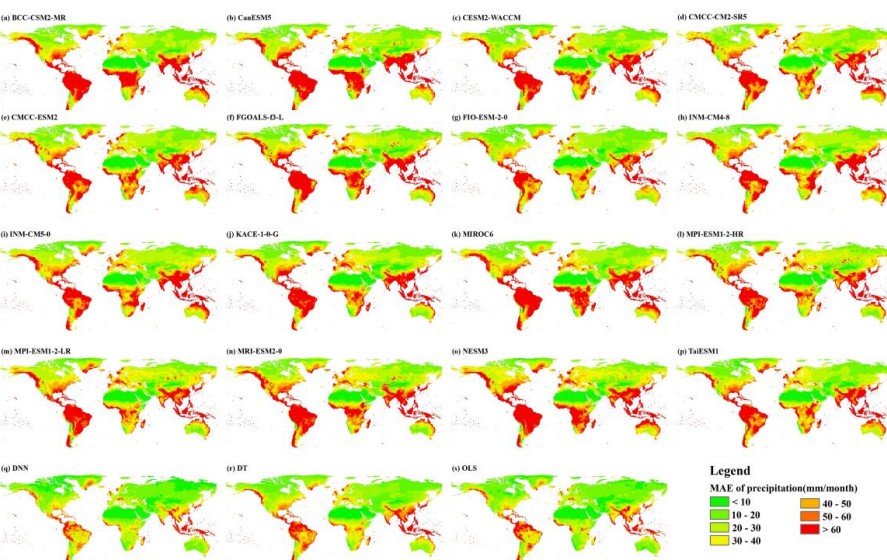

Fig. 6. The spatial distribution illustration of precipitation MAE produced by selected CMIP6
models, DNN (Deep Neural Networks), DT (Decision Tree) and OLS (Ordinary Least Squares
regression





3.1.3 Overall performance evaluation
Due to the partial model assessment of a single indicator, different metrices result in
different ranks. it is necessary to employ comprehensive index to improve the credible
evaluation. To further measure superiority of different models, different monthly index
rankings were calculated firstly before CRI assessment. The closer the pixel color to
green, the better the ranking is, vice versa. Each pixel in heatmaps of CRI ranking (Fig
6) was calculated by four indices (R, CRMSE, SD ratio and MAE) according to the
monthly ranking of single model and ensembled dataset. What cannot be ignored is that
the proposed datasets from three ML methods ranked ahead of CRI generated by four
indicators with green covered ribbons in both temperature (Fig 7.a) and precipitation
prediction (Fig 7.b). Particularly, the DNN method was the optimal one among
investigated single model and multi-model ensembled datasets. As for temperature, R
values for the DNN methods were all ranked first for all months. Results from the DNN
method ranked at 1 according to the CRMSE and MAE in each month except February,
in which it ranked at 2.

The precipitation dataset from the DNN method ranked 1 in all months according to the
MAE. The ranks with indicator R and CRMSE were either first or second indicating
stable and perfect performance of DNN. Based on the SD ratio, results from the DNN
method ranked middle. However, the SD ratio represented the overall pattern between
the observation and simulation instead of the corresponding relations sample by sample.




Therefore, the SD ratio was not regarded to be persuasive compare with other indicators.

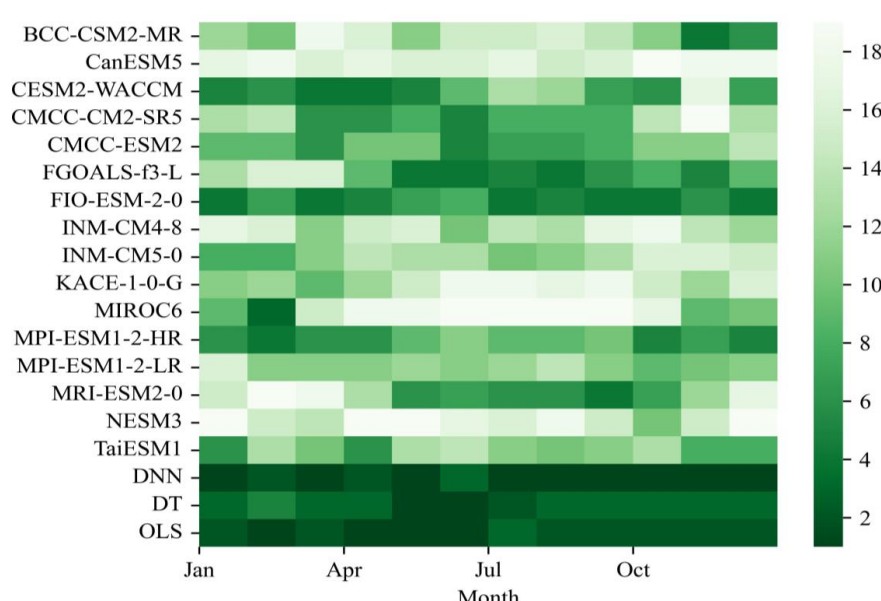





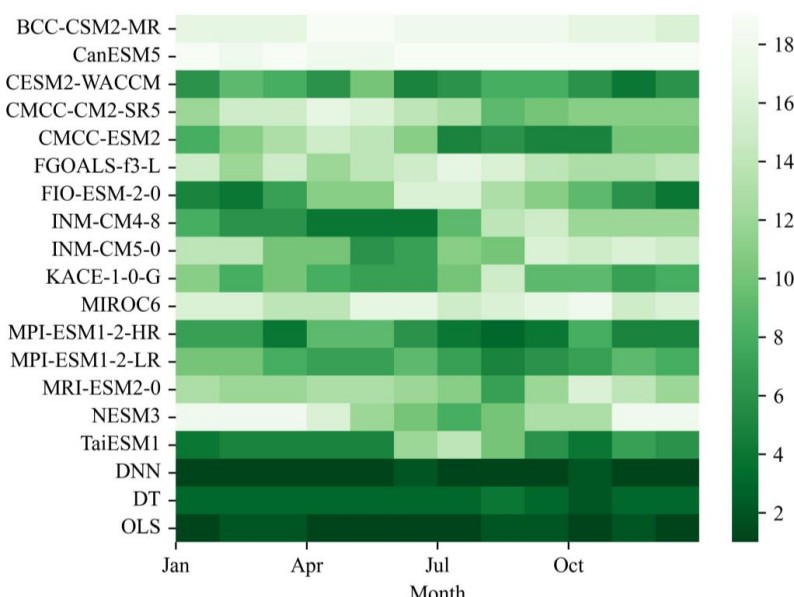

Fig. 7. CRI ranking of 16 single models and datasets from three ML methods. (a) temperature and (b) precipitation.

According to the CRI ranking results, the monthly optimal pattern was screened out to produce the new optimal dataset. In summary, the DNN method had an overwhelming advantage in all months except in February and April, in which the OLS method was the optimal method for temperature ensemble. On the other hand, the OLS was the best method for projecting precipitation from March to June and October, meanwhile the DNN produced optimal results in other months. Notably, there were two or more optimal methods in certain months (e.g., March, May) due to the same CRI ranking produced by the discrepancy of the partial indicator. Considering the stability, robustness, and R representing fitting ratio, the DNN method was employed as the optimal method for further predictive analysis when facing above situation.



*3.2 Years projection for temperature increasing under the 1.5℃ (2℃ / 3℃) global*
*warming target*
From the proposed optimal monthly dataset, temperature was projected under SSP1-
2.6, SSP2-4.5 and SSP5-8.5 scenarios for the period of 2015-2100. As well, the pre-
industrial period (1850-1900) dataset from CMIP6 was selected as reference to years
projection for temperature increasing under the 1.5℃ (2℃ / 3℃) global warming target.
For further intuitive analysis of temperature anomalies, global studied area was divided
into Asia, Africa, Europe, South America and North America and Oceania continents.
The temperature trends were shown in Figure 8. Clearly, the upward trend of SSP1-2.6
was steadier while steepest upward trend of the SSP5-8.5. What's more, Asia, Europe
and North America continents contributed more to global warming than Oceania, Africa
and South America continents in both scenarios.

The following simulated data are processed by 5-year moving average. In order to
further confirm the time period of temperature rise in the study area, the rising targets
of 1.5 ℃, 2 ℃ and 3 ℃ were set in Figure 8. Under the SSP1-2.6 scenario, Asia, Africa,
South America, Oceania and global reach 1.5 ℃ threshold in the year of 2031, 2050,
2034, 2072 and 2037, respectively. Europe and North America continents get to 2℃
rising level during 2027 to 2029. If future followed the medium emission scenario
namely SSP2-4.5, the years for Africa, South America and Oceania continents
breakthrough 1.5 ℃ (2℃ / 3℃) warming target were 2024 (2037/2075), 2026
(2043/2082) and 2029 (2038/2094). Asia reached 3 ℃ warming target in 2026-2031





and Europe reached 2 ℃ (3 ℃) level in 2026 (2040). Asia will firstly reach the 3 ℃
warming level, while Oceania continent is last one. The time breakpoints exceeding
1.5 ℃, 2 ℃ and 3 ℃ thresholds were 2029, 2035 and 2058 under the SSP2-4.5 scenario
in global scale. the SSP5-8.5 scenario was denoted fossil-fueled development
socioeconomic pathway. Therefore, it is not surprised to find the severity of temperature
rising is greater than SSP 2-4.5 scenario. Under the SSP5-8.5 scenario, the time periods
for global continent breakthrough 1.5 ℃, 2 ℃ and 3 ℃ rising threshold were 2024,
2031 and 2048, respectively. The period for Asia, Africa, Europe, South America and
North America and Oceania continents for 3 ℃ warming target were 2024, 2055, 2036,
2031, 2060 and 2062 under the SSP5-8.5.

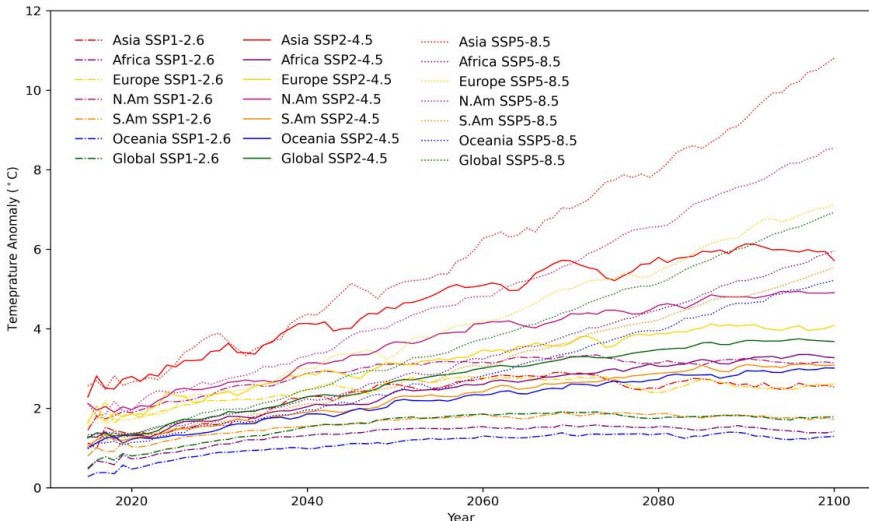

Fig. 8. Temperature anomalies of global and continents under (a) SSP1-2.6 (b) SSP2-
4.5 and (c) SSP5-8.5 respect to pre-industrial temperature (1850-1900). N. Am
denotes North America. S. Am denotes South America.

*3.3 Tracking global and continental future precipitation changes*
Monthly precipitation projection from 2015-2100 under three main scenarios were



analyzed in Fig. 9 and Fig. 10. As the color bar shown, the closer color of the cell is
bright red, the ampler the precipitation is. On the contrary, the closer the color is to
green, the absent the precipitation is. In this study, we defined the spring (March to
May), summer (June to August), Fall (September to November) and Winter (December
to next February) in both north and south hemispheres to facilitate consistent analysis
for different climate zones.

The tendency in intra-annual precipitation keeps rising under SSPs except for the
decreasing tendency of winter under SSP1-2.6 (Fig. 9). From 2020-2100, July and
August can be classified as wet months. On the other hand, April and September to next
February can be categorized as dry months. In detail, summer rainfall is the most
abundant. The amounts of summer value account for 31.6%, 29.1% and 29.8% of
annual rainfall with the increase rates of summer at 0.30 mm/10a, 0.16 mm/10a and
0.76 mm/10a under SSP1-2.6, SSP2-4.5 and SSP5-8.5. Although the monthly
precipitation in summer rank first in three selected scenarios, the increased monthly
rainfall slopes of autumn, which can be determined as the peak among above SSPs, are
0.28 mm/10a, 0.63 mm/10a and 1.418 mm/10a under SSP1-2.6, SSP2-4.5 and SSP5-
8.5, respectively. In terms of SSPs, the monthly wetter tendency of SSP5-8.5 is the most
significant with a rate of 1.14 mm/10a. However, it doesn't mean that more uniform
global precipitation distribution in all continents will happen.

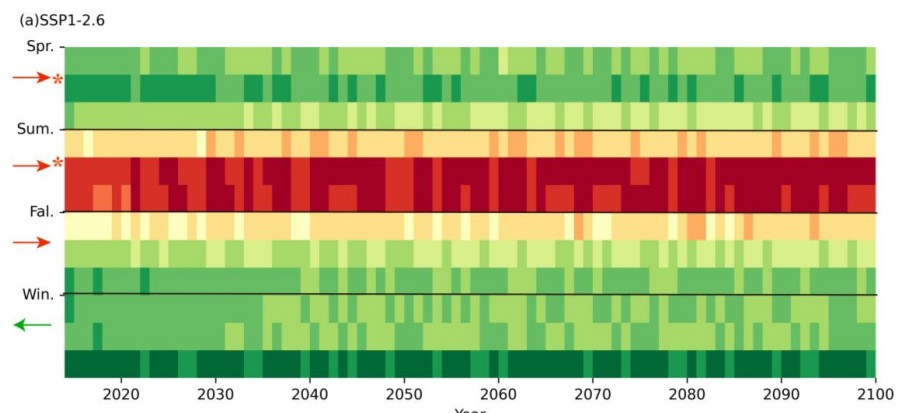


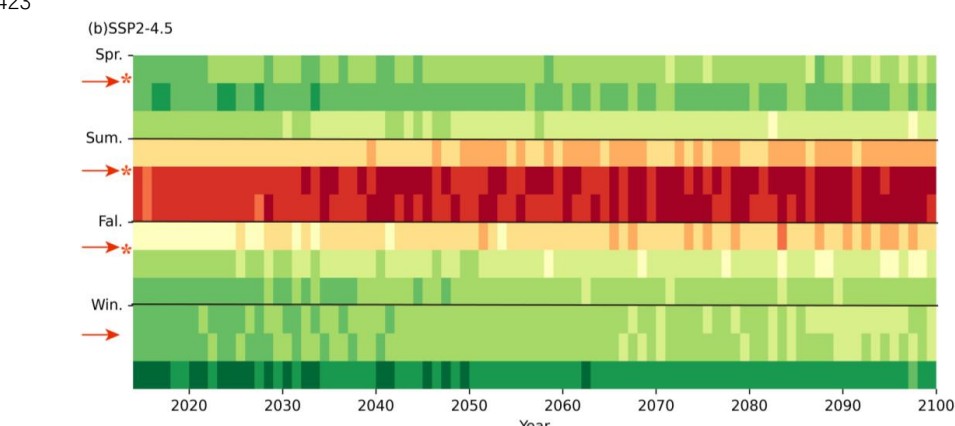


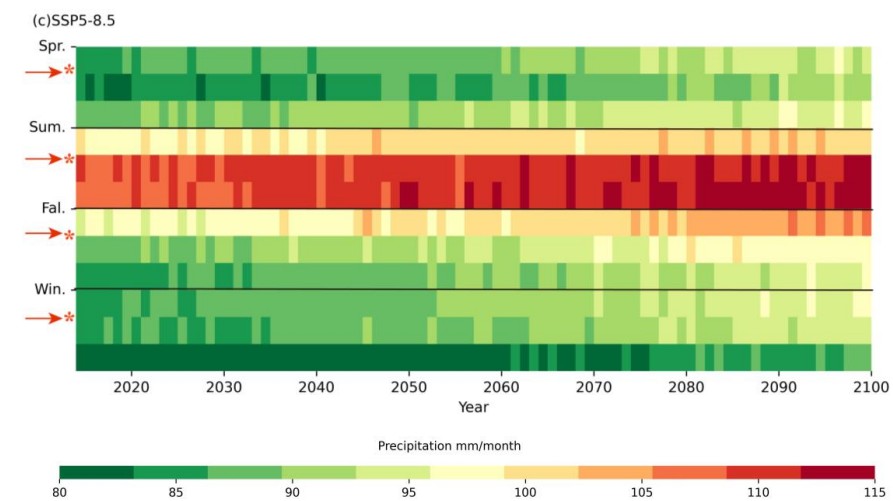



Fig. 9. Mean precipitation changes of each month for global continents under (a)
SSP1-2.6, (b) SSP2-4.5 and (c) SSP5-8.5. Each cell represents monthly mean
precipitation. Each row is sorted by spring (March to May), summer (June to August),





fall (September to October) and winter (December to February). The green arrow
turning left denotes downward trend, while red arrow facing right denotes upward
tendency. Asterisk represents significance value with $p<0.05$.

According to the abundance of precipitation, South America can be categorized as the
extremely rainy continent (Fig. 10a), while other studied continents can be grouped as
normally rainy continents (Fig. 10b). In respect of SSP1-2.6 and SSP2-4.5, all studied
continents exhibit increasing trends of monthly precipitation. While the largest
decreasing trend polarization of uneven precipitation at the continental scale under
SSP5-8.5 was further detected, suggesting SSP5-8.5 may cause floods or droughts. Asia
and Africa which can be classified as precipitation-deficit continents tend to be drier
from 2015-2100($p < 0.05$) with 19.7% and 15.2% decreasing trends. What's more,
South America will be more humid with as the most abundant precipitation continent.
Similarly, Europe and North America with relatively abundant precipitation will also
usher in more precipitation under SSP5-8.5. To assess the wetting trend of continents
more intuitively, the precipitation increases by 7.62%, 15.5% and 6.72% in Europe,
North America and South America continents, respectively, while the upward trend is
not obvious in Oceania continent.

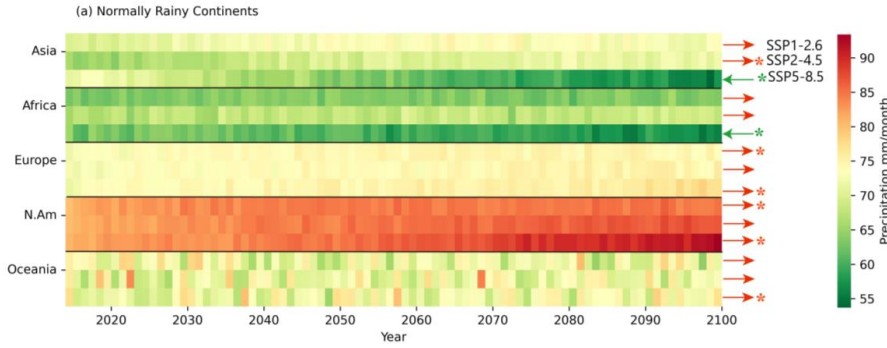


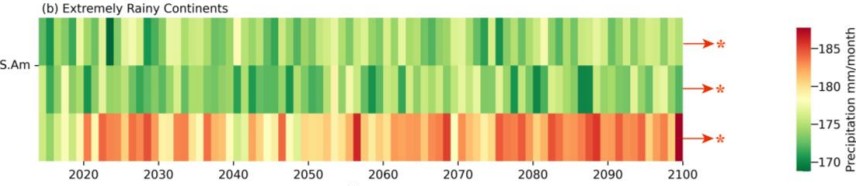


Fig. 10. Land mean rainfall changes of (a) normally rainy continents (Asia, Africa, Europe, N. Am (North America) and Oceania) and (b) extremely rainy continent: S. Am (South America). Each cell represents a monthly mean precipitation value of the continent land. The order of rows is SSP1-2.6, SSP2-4.5 and SSP5-8.5 for each continent. The green arrow turning left denotes downward trend, while red arrow facing right denotes upward tendency. Asterisk (*) represents significance value with $p<0.05$

**4. Discussion**

*4.1 Higher credibility of the proposed ensemble dataset by comparison with previous studies*

Majority of previous studies were based on CMIP5 to predict future temperature and precipitation for evaluating ecological impacts of climatic dynamics (Miao et al., 2014; Navarro-Racines et al., 2020; Putra et al., 2020; Kajtar et al., 2021;Tang et al., 2021; Wu et al., 2021). More skillful dataset can improve the spatial correlation accuracy and reduce the bias over the studied region. CMIP6 GCMs with higher resolutions and human activity simulation conditions have been proved with better performance in characterizing the completion processes of carbon emissions, radiative forcing and warming projection (Xin et al., 2020; Zamani et al., 2020; McCrystall et al., 2021; Song et al., 2021). The newly released CMIP6 GCMs were selected to simulate in this study. Besides the new alternation of data sources, there is further improvement of ensemble methods. To decrease the discrepancy between simulation and observation for higher



accuracy, traditional methods (e.g., multi-model ensemble mean, best fitting single
model selection) were applied (Rivera and Arnould., 2020; Baker et al., 2021; Kajtar et
al., 2021). It is noteworthy that traditional procedure lacks flexibility and ignores the
weight allocation of time dimensions. Studies have demonstrated that deep learning can
reproduce data in pattern coupling with excellent performance (Sun and Archibald,.
2021; Wei et al., 2021). In this study, considering temporal variation, the application of
neural network and machine learning reproduce dataset with higher ability of projecting
climatological rainfall and temperature under SSP1-2.6, SSP2-4.5 and SSP5-8.5.
Detailed assessment was conducted to find that three new methods are more faultless
than any single model. In terms of temperature (precipitation), MAE of proposed
dataset reduced from 4.4 °C (46.6 mm/month) to 2.1 °C (27.3 mm/month) compared
with single GCM data.

*4.2 Aggravation of global warming and precipitation extreme by socio-economic*
*pathways*
The RCP scenarios adopted in CMIP5 were labelled for the range of radiative forcing
values until 2100 (2.6, 4.5, 6, and 8.5 W·m$^{-2}$, respectively) (Rao and Garfinkel 2021).
However, SSP-RCPs are joined to describe national policies besides radiative forcing
during CMIP6 (Liao et al. 2020). There are different results of global warming and
precipitation extreme from these two phases, in which it seems more aggravative in
CMIP6 than CMIP5 according to the results from this study. Torres et al. (2022)
projected temperature for South America and stated that the years related to 1.5 °C and



2 °C thresholds were 2027 and 2040 under RCP8.5, while 2023 and 2034 under SSP5-
8.5 during CMIP6, respectively in this study, in which temperature increasing quicker
in CMIP5 than CMIP6. Additionally, Bokhari et al. (2018) claimed that the mean
temperature over South Asia showed an estimated temperature rising of 3.2°C under
RCP4.5 until 2050. Compared with the projection conducted by Bokhari et al. (2018),
we have noted that Asia will experience an increasing of 4.32 °C under RCP4.5, which
is more intensive than the tendency under SSP2-4.5 in the mid-21st century. Moreover,
Ongoma et al. (2018) estimated an increasing in temperature at 2.8 °C and 5.4 °C over
East Africa under the RCP4.5 and RCP8.5 scenarios until 2100, respectively. Notably,
the increasing tendency over Africa in CMIP6 of this study is 3.4 and 6.0 °C under
SSP2-4.5 and 5-8.5, respectively, which is acuter than the increment under RCP4.5 and
RCP8.5. Thus, global warming seems to be accelerated under the new socio-economic
pathways in CMIP6.

In terms of precipitation, Zhu et al. (2021) demonstrated that the annual precipitation
over China would increase by 4.4% and 7% in CMIP5, which is weaker than the trends
representing 5.3% and 8.6% under corresponding scenarios in CMIP6. Moreover, Sinha
et al. (2018) reported the precipitation Florida may experience 5% rising under RCP4.5,
which is 3% lower than trends in SSP2-4.5. It can be demonstrated that the changes of
temperature rising and precipitation extreme in these studies agree with our findings,
which reveals socio-economic pathways could aggravate global warming and
precipitation extreme in the 21[st] centry.




*4.3 Implication for climate changing pattern projected from proposed datasets*

It is obvious that the severity of climate changes follows the order of SSP5-8.5 > SSP2-4.5 > SSP1-2.6, in which the scenarios represent durable sustainability, intermediate and fossil-fuel driven high emissions, respectively. Under SSP5-8.5 scenario, GDP growth develops at high speed at the cost of high energy intension in the absence of newly proposed climate management policies. Compared with SSP1-2.6 and SSP2-4.5, time periods breakthrough warming targets come in advance under SSP5-8.5. The analysis results imply that we must adopt reasonable climate intervention policies, including through the pursuit of alternative clean energy instead of fossil fuel-driven approaches. This study also indicated that the phenomena that wet regions become wetter while dry regions become drier due to high emissions, is affected by economic development model to a certain extent. Therefore, conversion of economic development patterns is also one of the factors to be considered in drought and flood mitigation measures. In multi-propose ecological projects, hydropower, agricultural irrigation, drought monitoring and land utilization management need credible evaporation evidence (Paredes et al., 2020). The meteorological factors are related to evaporation estimation. (Lu et al., 2021; Tian et al., 2022). Related equations or indexes (e.g., Penman–Monteith, standardized precipitation index and the standardized precipitation evapotranspiration index) can be constructed employing climate variables to project future ecological system changes (Almorox et al., 2018; Pei et al., 2020). The



new ensemble climate dataset is expected to accurately project climate change and its
long-term effects of ecology and environment at a global scale.
**5. Conclusion**
In this study, high credible findings were proposed based on new ensemble CMIP6
ensemble dataset. We applied three machine learning methods (OLS, DT and DNN) to
construct new temperature and precipitation projection dataset, simultaneously. After
accuracy evaluation, the optimal monthly methods were selected to generate ensemble
dataset under SSP1-2.6, SSP2-4.5 and SSP5-8.5 scenarios. The optimal dataset proved
to be higher accuracy from five statistic indicators (R, CRMSE, MAE, SD ratio and
CRI) than CMIP6 single model. The ensemble dataset owned CRI ranking first and SD
ratio closing to 1 in each month. The new temperature dataset displayed perfect
simulation ($R = 0.99$, CRMSE = 0.19 °C, MAE = 2.05 °C) compared with single CMIP6
GCM ($0.95 < R < 0.97$, 0.25 °C< CRMSE < 0.30 °C, 3.45 °C < MAE < 4.39 °C), while
the new ensembled precipitation dataset was higher credible ($R = 0.81$, CRMSE = 0.61
mm/month, MAE = 27.31 mm/month) than the single CMIP6 GCM ($0.59 < R < 0.77$,
0.86 mm/month < CRMSE < 1.1 mm/month, 39.7 mm/month < MAE < 46.57
mm/month).

High credibility findings were conducted depending on this new dataset. Firstly, the
intensity order of temperature rising is SSP5-8.5 > SSP2-4.5 > SSP1-2.6 over a global
scale. Aisa, Europe and North America continents contributed more to global warming
than Oceania, Africa and South America continents under studied three SSPs scenarios.
Secondly, the global continent breakthrough 1.5 °C, 2 °C and 3 °C rising thresholds in
2024, 2031 and 2048, under SSP5-8.5 scenario. Thirdly, precipitation aggregated
during July and August over the global region. April and September to subsequent
February can be categorized as dry months under selected SSPs. Fourthly, the
ensembled dataset implicates that SSP5-8.5 scenario will accelerate global precipitation
polarization ($p < 0.05$). Precipitation changes in Africa and Asia will decrease,
meanwhile, Europe, Oceania and South America will be wetter under the SSP5-8.5
scenario. Associated with former studies, our findings proved that socio-economic
pathways could boost global warming and precipitation extreme.
**6. Data availability**
The CMIP6 GCMs can be downloaded at https://esgf-node.llnl.gov /search/cmip6/.
CRU TS4.05 dataset is available at https://crudata.uea.ac.uk/cru/data/hrg/cru_ts_4.05/.
The ensemble global new dataset can be accessed via open community Zenodo at
https://doi.org/10.5281/zenodo.6565574 (Lu and Zhang, 2022).

**Acknowledgments**
This work was funded by the National Key Research and Development Program
(2018YFC1506506), the Frontier Project of Applied Foundation of Wuhan
(2019020701011502), Key Research and Development Program of Jiangxi Province



(20201BBG71002), the ESA-MOST Dragon Program, and the LIESMARS Special
Research Funding.
**Conflict of interest**
The authors declared that there is no conflict of interest.

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
