# Peer review of "Machine-learning ensembled CMIP6 projection reveals socio-economic pathways"

_Hydrology and Earth System Sciences, 2022_

## Referee Comment (RC1)

1、 The authors should emphasize this work's main novelty or contribution more clearly.

2、 Though ML methods were successfully applied in the precious regional studies, regionalized models were just suitable for specified periods or regions. How can the authors promise the suitability of the ML methods at the global scale?

3、 By using the OLS model, did each GCM represent a dimension of space? How to set the weights for each GCM?

4、 Does the sum of $\beta_1^1, \beta_2^1, \ldots, \beta_{16}^1$ equal 1?

5、 Please make the code of this paper available for authors if possible.

6、 Different GCMs have different spatial resolutions; how do you simultaneously set them as the input? Did the authors downscale the GCMs to the same spatial resolution?

7、 The authors should improve image resolution.

There are also some tiny mistakes:

1) In Figure 1, the 30 CRU images in February should be changed to 196502

2) In line 165, "The process can be described as four steps in detail:" but only three steps are presented.

3) In line 173, it should be equation 5.

---

## Author Comment (AC1)

**Response to Referee #2,**

The comments of reviewers are posted in black; our answers are posted in blue.

**1. GENERAL COMMENTS**

The manuscript examined three ensemble mean methods, which are based on machine learning (ML), using 16 CMIP6 models and observed temperature and precipitation data. Their evaluation results suggested that three ML-based ensemble methods can generate more reasonable simulations relative to individual CMIP6 models. The Deep Neural Networks method shows an overall better performance than the Ordinary Least Squares regression and Decision Tree methods in terms of the ensemble mean precipitation and temperature of CMIP6 models. The authors also projected the changes in temperature and precipitation based on the proposed multi-model ensemble mean method for SSP126, SSP245, and SSP585.

**We would like to thank Referee#2 for taking the time to read the manuscript.**

The ensemble mean methods presented in the manuscript are interesting and the study fit the scope of HESS.

**Thank you very much for your appreciation.**

However, the methods were not well explained and many important details are missing. In addition, comparing the ML-based ensemble mean against individual CMIP6 models cannot well define the merit of the method. At least, the uniformly weighted multi-model ensemble mean should be included in the comparison. The main findings of the study were not well summarized, either. Given these issues, substantial revision is required before the manuscript can be considered for publication with HESS.

We would like to thank Referee#2 for the valuable feedback that we have used to significantly improve the quality of our manuscript. As you are concerned, there are problems that need to be addressed. We have carefully considered all comments from the reviewers and revised our manuscript. The revised evaluation has included MME. We have rewritten the main findings in revision and completed the missing details. The manuscript has also been double-checked, and the typos and grammar errors we found have been corrected. We believe that our responses may tackle all concerns from the reviewers.

**2. Major comments**

Q1: The authors evaluated the performance of ensemble mean derived from three machine-learning (ML) methods as well as individual CMIP6 model performance against observations. They concluded that the ML-based ensemble-mean precipitation and temperature were more credible than that simulated by individual GCMs. Previous studies suggested that multi-model ensemble mean with the uniform weight (MME) also shows generally good performance than individual GCMs in terms of various variables

or indices (e.g., Massoud et al, 2019; Zhang et al, 2022). It is not enough to justify the merit of ML-based ensemble mean by simply comparing it with individual GCMs. I strongly suggest the authors include MME in the evaluation and compared it with the ML-based ensemble mean.

Thank you for the suggestions which help us improve our work. We have followed the advice in our revision. The related assessment has been added and figures have been redrawn. Overall, the accuracy assessment has proved that the Deep Neural Networks method still performs better than MME. In terms of temperature(precipitation), Rs value of DNN (Rtemperature = 0.985, Rprecipitation = 0.819) and OLS (Rtemperature=0.983, Rprecipitation=0.800) are both higher than MME (Rtemperature=0.973, Rprecipitation=0.759). Additionally, **CRMSE** value of DNN (CRMSEtemperature =0.171°C/month, CRMSEprecipitation=0.618mm/month) and OLS (CRMSEtemperature=0.181°C/month, CRMSEprecipitation=0.619mm/month) are both less than MME (CRMSEtemperature =0.232°C/month, CRMSEprecipitation=0.651mm/month) (Fig. 1). The R of DT in temperature(precipitation) is 0.972(0.718), which is slightly lower than MME. Moreover, the median MAE of DNN and OLS in temperature(precipitation) are 1.88 °C/month and 1.96°C/month (18.3 mm/month and 18.7 mm/month), which is reduced by 21.43% (33.0%) and 18.0% (31.5%) compared with MME (Fig. 2). The MAE maps of temperature(precipitation) showed MME methods reduce MAE value in the northern(southern) hemisphere compared with single GCM. However, MME is inferior to DNN and OLS according to the spatial distribution illustration of MAE in temperature(Fig. 3) and precipitation(Fig. 4). The overall assessment produced by CRI showed that the DNN method still had an overwhelming advantage in all months except in February and April (Fig. 5a), in which the OLS method the optimal method for temperature ensemble. On the other hand, the OLS was the best method for projecting precipitation from March to June and October (Fig. 5b), meanwhile, the DNN produced optimal results in other months (Fig. 5b).

---

## Author Comment (AC2)

Response to Referee #1,

The comments of reviewers are posted in **black**; our answers are posted in blue.

Q1: The authors should emphasize this work's main novelty or contribution more clearly.

Thank you for your kind advice which helps improve our work.

In this study, we applied three machine learning methods (OLS, DT and DNN) to produce ensembled datasets during the time period of 2015-2100 under three SSP scenarios (SSP1-2.6, SSP2-4.5 and SSP5-8.5), to project changes in temperature and precipitation over the global continent.

Previous studies mainly project future climate changes through single GCMs or multi-model ensemble (MME) datasets. MME is also the ensemble method that shows generally good performance in terms of various variables or indices (e.g., Massoud et al, 2019; Zhang et al, 2022). We compared the accuracy of single GCM, MME and our new ensemble datasets. The accuracy assessment showed that the credibility of new datasets has improved compared with single GCMs and MME.

Former studies mainly focus on regional climate changes (Gaitán et al. 2019; Lee et al. 2022). The optimized datasets were applied to capture future climate change at continental and global scales. The climate characteristics at the global and continental scales in the future were proposed as follows.

(1) Aisa, Europe and North America continents contributed more to global warming than Oceania, Africa and South America continents under the studied three SSPs scenarios.

(2) The slopes of Asia under three SSPs are the largest among six continents, which are 0.165 ℃·10a$^{-1}$、0.439 ℃·10a$^{-1}$ and 0.961 ℃·10a$^{-1}$ under SSP1-2.6, SSP2-4.5 and SSP5-8.5.

(3) Precipitation will aggregate during July and August over the global continent. April and September to subsequent February can be categorized as dry months under selected SSPs.

(4) SSP5-8.5 scenario will accelerate global precipitation polarization ($p < 0.05$).

Refer

Gaitán, E., Monjo, R., Pórtoles, J., et al. Projection of temperatures and heat and cold waves for Aragón (Spain) using a two-step statistical downscaling of CMIP5 model outputs. Science of The Total Environment, 2019, 650, 2778-279.

Massoud E C, Espinoza V, Guan B, et al. Global climate model ensemble approaches for future projections of atmospheric rivers. Earth's Future, 2019, 7(10): 1136-1151.

Lee, Y., Paek, J., Park, J.-S., et al. Changes in temperature and rainfall extremes across East Asia in the CMIP5 ensemble. Theoretical and Applied Climatology, 2022, 141, 143-15.

Zhang M Z, Xu Z, Han Y, et al. Evaluation of CMIP6 models toward dynamical downscaling over 14 CORDEX domains. Climate Dynamics, 2022, 1-15.

Q2: Though ML methods were successfully applied in the precious regional studies, regionalized models were just suitable for specified periods or regions. How can the authors promise the suitability of the ML methods at the global scale?

Thank you for your question. The climate projection in this study is at a global scale. The accuracy assessment is also based on the observed, GCM and ensemble datasets which contain climate information of the global continent. Due to the missing of the observed dataset in the future, we verified the accuracy of the models in the historical period. The training time period of ensemble models is1965-1994 and the validation time period is 1995-2014. Thackeray et al. (2022) demonstrate that there exists a relationship between the future increased occurrence of precipitation extremes aggregated over the globe and the observable change over recent decades. Moreover, Thackeray et al. (2022) also proposed that model errors in the historical climate can be used to inform the best estimate of future change. Many studies assess the accuracy of GCMs in reproducing observed temperatures in historical periods and then employed multi-model ensembles to simulate climate changes (Boberg et al., 2012; Aloysius et al., 2017; Jia et al., 2019; Shiru et al., 2020).

Refer

Boberg F, Christensen J H. Overestimation of Mediterranean summer temperature projections due to model deficiencies. Nature Climate Change, 2012, 2(6): 433-436.

Aloysius N, Saiers J. Simulated hydrologic response to projected changes in precipitation and temperature in the Congo River basin. Hydrology and Earth System Sciences, 2017, 21(8): 4115-4130.

Jia K, Ruan Y, Yang Y, et al. Assessment of CMIP5 GCM simulation performance for temperature projection in the Tibetan Plateau. Earth and Space Science, 2019, 6(12): 2362-2378.

Shiru M S, Chung E S, Shahid S, et al. GCM selection and temperature projection of Nigeria under different RCPs of the CMIP5 GCMS. Theoretical and Applied Climatology, 2020, 141(3): 1611-1627.

Thackeray C W, Hall A, Norris J, et al. Constraining the increased frequency of global precipitation extremes under warming. Nature Climate Change, 2022, 12(5): 441-448.

Q3: By using the OLS model, did each GCM represent a dimension of space? How to set the weights for each GCM?

Thank you for your questions. Each GCM represents a dimension of space in a certain month.

To calculate the weight of each GCM in the $i^{th}$ month, Residual standard error (RSS) needs to be minimized, which can be described as follow.

$$RSS^i = \sum_{s}^{k=1} \left( Y^{(i,k)'} - Y^{(i,k)} \right)^2 \tag{3}$$

where $Y^{(i,k)'}$ and $Y^{(i,k)}$ denote the values of single kth pixel value in the CRU TS4.5 image and the ensemble image according to the temporal-spatial correspondence, respectively; s which equals to 2022600(67420*30) denotes the sum of pixels of the CRU TS4.5 (or ensemble) images during the period of 1965-1994.

Take the first derivative of Eq. (3) equal to 0 after substituting $Y^{(i,k)}$ in Eq. (2) into Eq. (3). The weight in the $i^{th}$ month can be determined as follows.

$$
\begin{bmatrix} \beta_1^i \\ \vdots \\ \beta_{16}^i \end{bmatrix} = (X_i^T X_i)^{-1} X_i^T Y_i \tag{4-1}
$$

$$
X_i = \begin{bmatrix} pixel_1^{(1,1965,i)} & \cdots & pixel_{15}^{(1,1965,i)} & pixel_{16}^{(1,1965,i)} \\ \vdots & \ddots & \vdots & \vdots \\ pixel_1^{(67420,1965,i)} & \cdots & pixel_{15}^{(67420,1965,i)} & pixel_{16}^{(67420,1965,i)} \\ \vdots & \ddots & \vdots & \vdots \\ pixel_1^{(67420,1994,i)} & \cdots & pixel_{15}^{(67420,1994,i)} & pixel_{16}^{(67420,1994,i)} \end{bmatrix} \tag{4-2}
$$

$$
Y_i = \begin{bmatrix} y^{(1,1965,i)} \\ \vdots \\ y^{(67420,1965,i)} \\ \vdots \\ y^{(67420,1994,i)} \end{bmatrix} \tag{4-3}
$$

where $\beta_k^i$ denotes the weight of the kth model in the $i^{th}$ month; $pixel_k^{(p,q,i)}$ and $y_k^{(p,q,i)}$ denote the value of the $p^{th}$ pixel in the $q^{th}$ year from the image of the kth model and CRU TS4.5, respectively.

Q4: Does the sum of $\beta11$, $\beta21$, …, $\beta16$ equal 1?

Thank you for raising this question. Considering the overestimation of models, the sum of $\beta$s doesn't equal 1. Constraining the sum equal to 1 may lead to overestimation. Paul (2022) proposed that GCMs exaggerate the impacts of global warming. Chai. et al. (2022) demonstrated the projected precipitation increase in the future is overestimated by CMIP6 over Asia.

Refer

Voosen Paul. "Hot" climate models exaggerate Earth impacts. Science, 2022, 376

Chai, Y., Yue, Y., Slater, L.J. et al. Constrained CMIP6 projections indicate less warming and a slower increase in water availability across Asia. Nature Communication, 2022, 13, 4124.

Q5: Please make the code of this paper available for authors if possible.

Thank you for your kind advice. The code is available via https://doi.org/10.5281/zenodo.7104329.

Refer

Piaoyin, Zhang, Jianzhong Lu: Machine Learning-GCM-WHU (1.0). Zenodo [code], https://doi.org/10.5281/zenodo.7104329

Q6: Different GCMs have different spatial resolutions; how do you simultaneously set them as the input? Did the authors downscale the GCMs to the same spatial resolution?

Thank you for raising this point. All GCM simulations and CRU data are re-gridded into a common $0.5° \times 0.5°$ resolution by the bilinear interpolation, which is a widely used method (Aloysius et al. (2016); Abbasian et al. (2018); Thackeray et al. (2022)). We have added related details to the revision.

Refer

Aloysius N R, Sheffield J, Saiers J E, et al. Evaluation of historical and future simulations of precipitation and temperature in central Africa from CMIP5 climate models. Journal of Geophysical Research: Atmospheres, 2016, 121(1): 130-152.

Abbasian, M., Moghim, S. and Abrishamchi, A. Performance of the general circulation models in simulating temperature and precipitation over Iran. Theoretical and Applied Climatology, 2018, 135, 1465–1483.

Thackeray C W, Hall A, Norris J, et al. Constraining the increased frequency of global precipitation extremes under warming. Nature Climate Change, 2022, 12(5): 441-448.

Q7: The authors should improve image resolution.

Thank you for your advice which improves our work. We have improved the resolution to 600dpi in the revision.

There are also some tiny mistakes:

1) In Figure 1, the 30 CRU images in February should be changed to 196502

Thank you raising this typo. We have corrected it in the revision as follows:

[Figure]

Fig. 1. Weight assignment of 16 GCMs on a time scale

2) In line 165, "The process can be described as four steps in detail:" but only three steps are presented.

Thank you for raising this typo. This sentence has been modified.

3) In line 173, it should be equation 5.

Thanks. We have made corrections according to your comments.

---

## Author Comment (AC3)

Response to Referee #2,

The comments of reviewers are posted in **black**; our answers are posted in **blue**.

**1. GENERAL COMMENTS**

The manuscript examined three ensemble mean methods, which are based on machine learning (ML), using 16 CMIP6 models and observed temperature and precipitation data. Their evaluation results suggested that three ML-based ensemble methods can generate more reasonable simulations relative to individual CMIP6 models. The Deep Neural Networks method shows an overall better performance than the Ordinary Least Squares regression and Decision Tree methods in terms of the ensemble mean precipitation and temperature of CMIP6 models. The authors also projected the changes in temperature and precipitation based on the proposed multi-model ensemble mean method for SSP126, SSP245, and SSP585.

We would like to thank Referee#2 for taking the time to read the manuscript.

The ensemble mean methods presented in the manuscript are interesting and the study fit the scope of HESS.

Thank you very much for your appreciation.

However, the methods were not well explained and many important details are missing. In addition, comparing the ML-based ensemble mean against individual CMIP6 models cannot well define the merit of the method. At least, the uniformly weighted multi-model ensemble mean should be included in the comparison. The main findings of the study were not well summarized, either. Given these issues, substantial revision is required before the manuscript can be considered for publication with HESS.

We would like to thank Referee#2 for the valuable feedback that we have used to significantly improve the quality of our manuscript. As you are concerned, there are problems that need to be addressed. We have carefully considered all comments from the reviewers and revised our manuscript. The revised evaluation has included MME. We have rewritten the main findings in revision and completed the missing details. The manuscript has also been double-checked, and the typos and grammar errors we found have been corrected. We believe that our responses may tackle all concerns from the reviewers.

**2. Major comments**

Q1: The authors evaluated the performance of ensemble mean derived from three machine-learning (ML) methods as well as individual CMIP6 model performance against observations. They concluded that the ML-based ensemble-mean precipitation and temperature were more credible than that simulated by individual GCMs. Previous studies suggested that multi-model ensemble mean with the uniform weight (MME) also shows generally good performance than individual GCMs in terms of various variables

or indices (e.g., Massoud et al, 2019; Zhang et al, 2022). It is not enough to justify the merit of ML-based ensemble mean by simply comparing it with individual GCMs. I strongly suggest the authors include MME in the evaluation and compared it with the ML-based ensemble mean.

Thank you for the suggestions which help us improve our work. We have followed the advice in our revision. The related assessment has been added and figures have been redrawn. Overall, the accuracy assessment has proved that the Deep Neural Networks method still performs better than MME. In terms of temperature(precipitation), $Rs$ value of DNN ($R_{temperature}$ = 0.985, $R_{precipitation}$ = 0.819) and OLS ($R_{temperature}$=0.983, $R_{precipitation}$=0.800) are both higher than MME ($R_{temperature}$=0.973, $R_{precipitation}$=0.759). Additionally, CRMSE value of DNN ($CRMSE_{temperature}$ =0.171°C/month, $CRMSE_{precipitation}$=0.618mm/month) and OLS ($CRMSE_{temperature}$=0.181°C/month, $CRMSE_{precipitation}$=0.619mm/month) are both less than MME ($CRMSE_{temperature}$ =0.232°C/month, $CRMSE_{precipitation}$=0.651mm/month) (Fig. 1). The R of DT in temperature(precipitation) is 0.972(0.718), which is slightly lower than MME. Moreover, the median MAE of DNN and OLS in temperature(precipitation) are 1.88 °C/month and 1.96°C/month (18.3 mm/month and 18.7 mm/month), which is reduced by 21.43% (33.0%) and 18.0% (31.5 %) compared with MME (Fig. 2). The MAE maps of temperature(precipitation) showed MME methods reduce MAE value in the northern(southern) hemisphere compared with single GCM. However, MME is inferior to DNN and OLS according to the spatial distribution illustration of MAE in temperature(Fig. 3) and precipitation(Fig. 4). The overall assessment produced by CRI showed that the DNN method still had an overwhelming advantage in all months except in February and April (Fig. 5a), in which the OLS method the optimal method for temperature ensemble.  On the other hand, the OLS was the best method for projecting precipitation from March to June and October (Fig. 5b), meanwhile, the DNN produced optimal results in other months (Fig. 5b).

[Figure]

Fig. 1. Taylor diagrams of (a) temperature and (b) precipitation. Ref stands for CRU TS4.05 observation dataset. SD and CRMSE were normalized by observed SD.

[Figure]

Fig. 2. Boxplots of Quantitative MAE assessment between simulation and observation dataset for (a) temperature (℃) and (b) precipitation (mm/month). The statistical distribution of data was displayed based on a five-divided category (minimum, first quartile, median, third percentile and maximum).

[Figure]

Fig. 3. The spatial distribution illustration of temperature MAE produced by selected CMIP6 models, DNN (Deep Neural Networks), DT (Decision Tree), OLS (Ordinary Least Squares regression) and MME.

[Figure]

Fig. 4. The spatial distribution illustration of precipitation MAE produced by selected CMIP6 models, DNN (Deep Neural Networks), DT (Decision Tree), OLS (Ordinary Least Squares regression) and MME.

[Figure]

(a)Temperature

[Figure]

(b)Precipitation

Fig. 5. CRI ranking of 16 single models and datasets from three ML methods. (a) temperature and (b) precipitation.

Q2: The methods were not explained in enough details. For example, how was the weight determined in Eq. (1)? What does p mean in Eq. (2)? Does the OLS method require a significant correlation between

model and observation? Note that no significant correlation is very common if one compares the year-to-year variation of precipitation (or temperature) between CMIP6 output and observation. What does "s" refer to in Eq. (3)? How was the weight determined in Eq. (7)? Without detailed method description, it is hard for readers to follow the authors' method and repeat their results.

Thank you for raising these points. We have prepared a revised manuscript that provides a more detailed description of the machine learning methods.

1)In order to better explain the principle of weight assignment, we explain it after pointing out the weight application (Eq. (2)).

The revised paragraph in the **Section 2.2** can be read as (the yellow highlights are added text):

To obtain ensemble value of each pixel, the linear model generated by OLS can be described as follow.

$$Y^{(i,k)} = \sum_{p}^{j=1} \beta_j^i X_j^{(i,k)} + \varepsilon_i \tag{2}$$

where $Y^{(i,k)}$ and $X_j^{(i,k)}$ denote the values of single $k^{th}$ pixel value in the ensemble image and the image of $j^{th}$ GCM in the $i^{th}$ month, respectively; $p$ equal to16 stands for the number of CMIP6 single model; $\varepsilon_i$ denotes the same meaning as in Equation (1).

To calculate the weight of each GCM in the $i^{th}$ month, Residual standard error (RSS) needs to be minimized, which can be described as follow.

$$RSS^i = \sum_{s}^{k=1} \left(Y^{(i,k)'} - Y^{(i,k)}\right)^2 \tag{3}$$

where $Y^{(i,k)'}$ and $Y^{(i,k)}$ denote the values of single $k^{th}$ pixel value in the CRU TS4.5 image and the ensemble image according to the temporal-spatial correspondence, respectively; $s$ which equals to 2022600 denotes the sum of pixels of the CRU TS4.5 (or ensemble) images during the period of 1965-1994.

Take the first derivative of Eq. (3) equal to 0 after substituting $Y^{(i,k)}$ in Eq. (2) into Eq. (3). The weight in the $i^{th}$ month can be determined as follows.

$$\begin{bmatrix} \beta_1^i \\ \vdots \\ \beta_{16}^i \end{bmatrix} = (X_i^T X_i)^{-1} X_i^T Y_i \tag{4-1}$$

$$X_i = \begin{bmatrix} pixel_1^{(1,1965,i)} & \cdots & pixel_{15}^{(1,1965,i)} & pixel_{16}^{(1,1965,i)} \\ \vdots & \ddots & \vdots & \vdots \\ pixel_1^{(67420,1965,i)} & \cdots & pixel_{15}^{(67420,1965,i)} & pixel_{16}^{(67420,1965,i)} \\ \vdots & \ddots & \vdots & \vdots \\ pixel_1^{(67420,1994,i)} & \cdots & pixel_{15}^{(67420,1994,i)} & pixel_{16}^{(67420,1994,i)} \end{bmatrix} \qquad (4-2)$$

$$Y_i = \begin{bmatrix} y^{(1,1965,i)} \\ \vdots \\ y^{(67420,1965,i)} \\ \vdots \\ y^{(67420,1994,i)} \end{bmatrix} \qquad (4-3)$$

where $\beta_k^i$ denotes the weight of the $k^{th}$ model in the $i^{th}$ month; $pixel_k^{(p,q,i)}$ and $y_k^{(p,q,i)}$ denote the value of the $p^{th}$ pixel in the $q^{th}$ year from the image of the $k^{th}$ model and CRU TS4.5, respectively.

2) Thank you for raising this point. $p$ equal to16 stands for the number of CMIP6 single model. We have clarified $p$ in the revision.

3) Thank you for raising this point. OLS method doesn't require a significant correlation between model and observation. In the revision, we have noted that significant correlation is not obvious generally when year-to-year variation between CMIP6 output and observation is compared.

4) Thank you for raising this point. "s" refer to the split node which was denotes the average value between the consecutive points in the same GCM after sorting the pixel values. The revised paragraph in the **Section 2.2** reads as (the yellow highlights are added or modified text):

The process of decision tree can be described as three steps in details:

Step1: Splitting the pixel values into two groups via the split node $s$ and split dimension $j$. Each GCM represents a dimension of a space. The split node denotes the average value between the consecutive points in the same GCM after sorting the pixel values. Select random value to initialize $j$ and $s$. Dividing the $j^{th}$ dimension of the space into two regions (R1 and R2) by selected candidate splitting the $j^{th}$ GCM as the feature, and then splitting the pixel values into two groups as following equations.

$$R1(j, s) = \{x \mid x(j) \le s\} \qquad (5-1)$$

$$R2(j, s) = \{x \mid x(j) > s\} \qquad (5-2)$$

Step2: Adjusting the $j$ and $s$ to minimize the residual sum of squares following equations.

$$min_{j,s}\left[ min_{c_1} \sum_{x_i \in R_1(j,s)} (y_i - c_1)^2 + min_{c_2} \sum_{x_i \in R_2(j,s)} (y_i - c_2)^2 \right] \qquad (6-1)$$

$$c_m = \frac{1}{N_m} \sum_{y_i \in R_1(j,s)} y_i \ (x \in R_m, m = 1,2) \tag{6-2}$$

where $N_m$ is the total number (30 images × 67420 pixels/image) of observation data at current node; $y_i$ is the $i^{th}$ individual sample of observation data.

Step 3: Repeating steps 1 and 2 to continue increasing the depth of tree and splitting the subregions R1 and R2 until training loss reaches to criteria threshold. Mean-absolute-error was applied as supported criteria to measure the quality of a split in this study.

5)Thank you for your question. The neural network was designed (Fig. 1) with 1024 batch size and 0.001 learning rate.

[Figure]

Fig. 1. Main Deep Neural Networks structure constructed in study. $\omega^l_{j,k}$ represents the weight from the $j^{th}$ neuron in the $(l+1)^{th}$ layer to the $k^{th}$ neuron in the $l^{th}$ layer.

There are 2022600 samples (30 images × 67420 pixels/image). Supposing there are m and n neurons in the $k^{th}$ and $(k+1)^{th}$ layers, respectively, the output weight $a^k$ of the $k^{th}$ layer in the $e^{th}$ epoch can be described as follow.

$$a^{(k+1,e)} = ReLU(W^{(k,e)}a^{(k,e)} + b^{(k,e)}) \tag{7-1}$$

$$ReLU(x) = \max(0, x) \tag{7-2}$$

where $b^k$ represents n×1 residual vector; $W^k$ represents a n×m weight matrix composed of linear coefficient of the $k^{th}$ layer. Note that $b^k$ was added into each column of $W^k a^k$.

The process of weight determined for the $i^{th}$ month can be described as follows:

(1) When epoch equals to 1, $W^{(1,1)}$ and $b^{(1,1)}$ were initialized to zero matrix ($1024 \times 16$ dimensions) and vector ($16 \times 1$ dimensions) respectively. $a^{(1,1)}$ represent the matrix($16 \times 1024$ dimensions) which can be described as follows:

$$a^{(1,1)} = [sample_1, \dots, sample_{1024}] \tag{8-1}$$

$$sample_1 = \left[pixel_1^{(y,k,i)} \dots pixel_{16}^{(y,k,i)}\right]^T \tag{8-2}$$

$$z^{(k,e)} = W^{(k,e)} a^{(k,e)} + b^{(k,e)} \tag{8-3}$$

where $pixel_m^{(y,k,i)}$ denote the value of the $k^{th}$ pixel in the $y^{th}$ year from the image of the $m^{th}$ model; $b^{(k,e)}$ represents $n \times 1$ residual vector for the $k^{th}$ layer in the $e^{th}$ epoch; $W^{(k,e)}$ represents a $n \times m$ weight matrix composed of linear coefficient of the $k^{th}$ layer; $z^k$ represents the temporary vector which need to be activated.

(2) To get more nonlinear and effective features, ReLU function was applied as activation in this paper.

$$a^{(k+1,e)} = ReLU\left(z^{(k,e)}\right) = max\left(0, z^{(k,e)}\right) \tag{9}$$

(3) Repeat step1 and step2 until k equals to 5 which denotes the forward propagation has been stopped.

(4) Start the process of back propagation. The current gradients $\nabla_{W^{(k,e)}}$ and $\nabla_{b^{(k,e)}}$ of $W^{(k,e)}$ and $b^{(k,e)}$ were calculated by taking partial derivatives with the loss function. The equation of $\nabla_{W^{(k,e)}}$, $\nabla_{b^{(k,e)}}$ and loss function can be described as follows.

$$\nabla_{W^{(k,e)}} = \frac{d_{LOSS\left(a^{(last,e)}, a^{e\prime}\right)}}{d_{W^{(k,e)}}} \tag{10-1}$$

$$\nabla_{b^{(k,e)}} = \frac{d_{LOSS\left(a^{(last,e)}, a^{e\prime}\right)}}{d_{b^{(k,e)}}} \tag{10-2}$$

$$LOSS\left(a^{(last,e)}, a^{e\prime}\right) = \frac{1}{n}\left\|\left(a^{(last,e)} - a^{e\prime}\right)\right\|^2 \tag{10-3}$$

where $a^{(last,e)}$ represent the ensemble vector ($1024 \times 1$ dimensions) of the input 1024 samples in the $e^{th}$ epoch; $a\prime$ represent the observation dataset of 1024 samples ($1024 \times 1$ dimensions)

(5) Update $W^{(k,e)}$ and $b^{(k,e)}$ in the opposite direction of the gradient until $\nabla_W$ and $\nabla_b$ of each layer have been calculated. The process can be described as follows.

$$W^{(k,e)} = W^{(k,e)} - lr * \nabla_{W^{(k,e)}} \qquad (11-1)$$

$$b^{(k,e)} = b^{(k,e)} - lr * \nabla_{b^{(k,e)}} \qquad (11-2)$$

where lr represent the learning rate which is set to 0.001.

(6) Repeat step 3 and step 4 until the k equal to 1 which denote back propagation is end.

(7) When epoch equal to 2, the steps towards step1. The $W^{(k,2)}$ and $b^{(k,2)}$ were initialized to updated $W^{(k,1)}$ and $b^{(k,1)}$. 1024 other samples are added to take the place of original 1024 samples in epoch1.

(8) Repeat the process of epoch1 until the training time reaches to 2000. At the same time, $LOSS\left(a^{(last,e)}, a^{e'}\right)$ needs to be monitored. Take the $W^{(k,e)}$ as the optimal weight when $LOSS\left(a^{(last,e)}, a^{e'}\right)$ is minimized.

Q3: The evaluation of three ML-based ensemble mean and individual CMIP6 models is not explained clearly. For example, it is not clear that the authors evaluated the climatological mean' precipitation (temperature) or their temporal variation against observation. It seems the authors evaluated the climatological means. Note that a model or multi-model ensemble mean can better capture the climatological means (section 3.1) do not necessarily suggest it can also generate a more reliable projection of future climate (section 3.2). No relationship between historical simulation and future projection was established in the evaluation, either. Moreover, the authors did not consider the uncertainty range of climate projection, which is also crucial for climate projection. These caveats should be discussed in the Discussion section at least.

Thank you for raising these points. CRU TS4.5 monthly dataset (1995-2014) was applied as the observation for accuracy evaluation. These indices represent the accuracy of GCMs and ensemble model dataset (1995-2014) against observation (CRU TS4.5) over global continent corresponding to the year-to-year temporal variation, respectively. There are 67420 pixels in each image which represent the global continent. For example, the R value of BCC-CSM2-MR in Taylor Diagram was calculated as follows:

(1) According to the year-to-year and temporal relationship, get the pairs of value of BCC-CSM2-MR and observed dataset in January. The time period was January of 1994-2015. CRU TS4.5 dataset (1994-2015) was applied as the observed dataset. Therefore, there are 1348400 pairs (20 years* 1 image/year* 67420 pixels/image) of GCM and observed values.

(2) Repeat step1, get pairs of values in other 11 months. The sum of pairs is 16180800(1348400 pairs/month *12 months).

(3) Calculate R through 16180800 pairs to represent the R value of BCC-CSM2-MR in Taylor diagram.

To calculate MAE, we get the values of $i^{th}$ (1≤i≤67420) location from GCM dataset and observed dataset (1995-2014) in each month corresponding to temporal relationship. Therefore, there are 240

pairs for the $i^{th}$ location to calculate the MAE of this pixel. For each GCM, the spatial pattern of MAE can be drawn as a map according to 67420 MAEs of pixels. Finally, we percentile of MAE of the $j^{th}$ GCM according to the 67420 values which represent the MAE of each spatial location.

We have clarified it in the revision. The assessment based on the temporal relationship proved the ensemble dataset may improve the credibility.

The method of calculating MAE, R, CRMSE and STD ratio is referred to previous studies (Deidda R et al., 2013; Wang B et al., 2018). The model we trained in historical was applied to project climate change. Due to the missing of future observed dataset, we verified model over recent decades. Thackeray et al (2022) utilized observational datasets to assess GCM performance in historical simulations to provide possible constraints on future projections. Moreover, Thackeray et al. (2022) also proposed that model errors in the historical climate can be used to inform the best estimate of future change. Many studies assess the accuracy of GCMs in reproducing observed temperatures in historical periods and then employed multi-model ensembles to simulate climate changes (Boberg et al., 2012; Aloysius et al., 2017; Jia et al., 2019; Shiru et al., 2020).

Refer

Aloysius N, Saiers J. Simulated hydrologic response to projected changes in precipitation and temperature in the Congo River basin. Hydrology and Earth System Sciences, 2017, 21(8): 4115-4130.

Boberg F, Christensen J H. Overestimation of Mediterranean summer temperature projections due to model deficiencies. Nature Climate Change, 2012, 2(6): 433-436.

Deidda R, Marrocu M, Caroletti G, et al. Regional climate models' performance in representing precipitation and temperature over selected Mediterranean areas. Hydrology and Earth System Sciences, 2013, 17(12): 5041-5059.

Jia K, Ruan Y, Yang Y, et al. Assessment of CMIP5 GCM simulation performance for temperature projection in the Tibetan Plateau. Earth and Space Science, 2019, 6(12): 2362-2378.

Shiru M S, Chung E S, Shahid S, et al. GCM selection and temperature projection of Nigeria under different RCPs of the CMIP5 GCMS. Theoretical and Applied Climatology, 2020, 141(3): 1611-1627.

Thackeray C W, Hall A, Norris J, et al. Constraining the increased frequency of global precipitation extremes under warming. Nature Climate Change, 2022, 12(5): 441-448.

Wang B, Zheng L, Liu D L, et al. Using multi-model ensembles of CMIP5 global climate models to

    reproduce observed monthly rainfall and temperature with machine learning methods in Australia.

    International Journal of Climatology, 2018, 38(13): 4891-4902.

We agreed with Reviewer #2's proposed caveats or limitations which have been discussed in the **section 4.1.** in the revision. The limitations can be read as follows:

However, there are some limitations of this work. Firstly, the observation data which has been applied is reanalysis data produced by information of meteorological stations. It exists the uncertainty which cannot been avoided and affect the accuracy. In the future, we will take the weather station data into consider to applied it with reanalysis data together. Secondly, the model uncertainty and internal variability will contribute to the uncertainty range (Samset et al., 2022) which is another meaningful research direction. We will consider explore the uncertainty range of ensemble dataset in the future.

Samset, B.H., Fuglestvedt, J.S., Lund, M.: Reply to: Uncertainty in near-term temperature evolution must not obscure assessments of climate mitigation benefits. Nature Communication,13, 4027 (2022). https://doi.org/10.1038/s41467-022-31426-w

Q4: The conclusion section did not well summarize the main findings of this study. Some conclusions are not new. For example, it is not a very new finding that CMIP6 models tend to project a warmer climate relative to CMIP5 models (Section 4.2). The "hot model" problem was also reported in recent studies (e.g. Hausfather et al, 2022; Voosen, 2022). Also, it is not surprising that "the intensity order of temperature rising is SSP5-8.5> SSP2-4.5> SSP1-2.6 over a global scale" (L553-555)

Thank you for raising this point. We raised the point that CMIP6 models tend to project a warmer climate relative to CMIP5 models in Discussion (Section 4.2), which is to revisit the views in former studies. This point in this study is in agreement with "hot model". We agree with your point that "the intensity order of temperature rising is SSP5-8.5> SSP2-4.5> SSP1-2.6 over a global scale" is common. Therefore, we have deleted L553-L555.

In this study, we applied three machine learning methods (OLS, DT and DNN) to produce ensembled datasets during the time period of 2015-2100 under three SSP scenarios (SSP1-2.6, SSP2-4.5 and SSP5-8.5), to project changes in temperature and precipitation over the global continent.

Previous studies mainly project future climate changes through single GCMs or multi-model ensemble (MME) datasets. MME is also the ensemble method that shows generally good performance in terms of various variables or indices (e.g., Massoud et al, 2019; Zhang et al, 2022). We compared the accuracy of single GCM, MME and our new ensemble datasets. The accuracy assessment showed that the credibility of new datasets has improved compared with single GCMs and MME.

Former studies mainly focus on regional climate changes (Gaitán et al. 2019; Lee et al. 2022). The optimized datasets were applied to capture future climate change at continental and global scales. The climate characteristics at the global and continental scales in the future were proposed as follows.

(1) Aisa, Europe and North America continents contributed more to global warming than Oceania, Africa and South America continents under the studied three SSPs scenarios.

(2) The slopes of Asia under three SSPs are the largest among six continents, which are 0.165 ℃·10a$^{-1}$、

0.439 ℃·10a$^{-1}$ and 0.961 ℃·10a$^{-1}$ under SSP1-2.6, SSP2-4.5 and SSP5-8.5.

(3) Precipitation will aggregate during July and August over the global continent. April and September to subsequent February can be categorized as dry months under selected SSPs.

(4) SSP5-8.5 scenario will accelerate global precipitation polarization (p < 0.05).

Refer

Gaitán, E., Monjo, R., Pórtoles, J., et al. Projection of temperatures and heat and cold waves for Aragón (Spain) using a two-step statistical downscaling of CMIP5 model outputs. Science of The Total Environment, 2019, 650, 2778-279.

Massoud E C, Espinoza V, Guan B, et al. Global climate model ensemble approaches for future projections of atmospheric rivers. Earth's Future, 2019, 7(10): 1136-1151.

Lee, Y., Paek, J., Park, J.-S., et al. Changes in temperature and rainfall extremes across East Asia in the CMIP5 ensemble. Theoretical and Applied Climatology, 2022, 141, 143-15.

Zhang M Z, Xu Z, Han Y, et al. Evaluation of CMIP6 models toward dynamical downscaling over 14 CORDEX domains. Climate Dynamics, 2022, 1-15.

**3. Minor comments**

Q5: L17: "precious studies", typo?

Thank you for catching this mistake. The text should be "previous studies" and we have corrected it in the revision.

Q6: L24,25: "The new ensemble precipitation (temperature) data with the R=0.81(0.99) is more accurate". It is not clear to me that the authors refer to climatological mean precipitation or interannual variation of precipitation. What data is used as the reference data when calculating the correlation coefficient?

In the process of calculating Taylor Diagram indices, we evaluated the accuracy of GCMs and ensemble model dataset (1995-2014) against observation corresponding to the year-to-year temporal variation. There are 67420 pixels in each image which represent the global continent. For example, the R value of BCC-CSM2-MR in Taylor Diagram was calculated as follows:

(1) According to the year-to-year and temporal relationship, get the pairs of value of BCC-CSM2-MR and observed dataset in January. The time period was January of 1994-2015. CRU TS4.5 dataset (1994-2015) was applied as the observed dataset. Therefore, there are 1348400 pairs (20 years* 1 image/year* 67420 pixels/image) of GCM and observed values.

(2) Repeat step1, get pairs of values in other 11 months. The sum of pairs is 16180800(1348400 pairs/month *12 months).

(3) Calculate R through 16180800 pairs to represent the R value of BCC-CSM2-MR in Taylor diagram.

The reference data is CRU TS4.05(1995-2014). The limitation of CRU TS4.05 has been added in revision, which can be read as "The observation data which has been applied is reanalysis data produced by information of meteorological stations. It exists the uncertainty which cannot been avoided and affect the accuracy. In the future, we will take the weather station data into consider to applied it with reanalysis data together".

We have clarified it in the revision. The assessment based on the temporal relationship proved the ensemble dataset may improve the credibility.

Q6: L32: "The proposed analysis provides credible opportunities…". Here and elsewhere, you may say "improve the credibility" rather than "credible projection of future climate". Given the large uncertainties, no climate projection is credible.

We have corrected these in the revision.

Q7: L45: "anthropogenic activities" -> "human activities"

We have corrected it in the revision.

Q8: L52: "GCMs (General Circulation Models)" -> "General Circulation Models (GCMs)"We have We corrected it in the revision.

Q9: L53: "catch" -> "project"

We have corrected it in the revision.

Q10: L63-64: "However, the findings generated by new ensemble climate global dataset are rarely reported under CMIP6 with the new emission strategy." What do you mean by "new ensemble climate global dataset". Many papers have been published in terms of climate projection with the CMIP6 dataset.

Thank you for raising this point. The method of producing new ensemble climate mainly focusing on MME. "new ensemble climate global dataset" denote the dataset ensembled by machine learning. We have modified it in the revision.

Q11: L67: "physical parameters sensitivity" is not the only factor that affects the model performance.

Thank you for raising this point. We have corrected it in the revision. Model performance will be affected by internal and external factors (e.g., misrepresentations of physical atmospheric, uncertainties regarding the boundary, initial model conditions and grid resolution).

Reference:

Murphy, J., Sexton, D., Barnett, D. et al. Quantification of modelling uncertainties in a large ensemble of climate change simulations. Nature, 430, 768–772, 2004.

Bromwich D H, Otieno F O, Hines K M, et al. Comprehensive evaluation of polar weather research and forecasting model performance in the Antarctic. Journal of Geophysical Research, Atmospheres, 118(2): 274-292,2013.

Papadimitriou, L. V., Koutroulis, A. G., Grillakis, M. G., et al. The effect of GCM biases on global runoff simulations of a land surface model, Hydrology Earth System Science, 21, 4379–4401, https://doi.org/10.5194/hess-21-4379-2017, 2017.

Q12:L68-69: "Climate change projection ignoring the temporal and spatial heterogeneity leads to the incredibility of the estimation." This is not true. Climate change projections do consider the temporal and spatial differences. Moreover, a projection of climate change at a global or continental scale is usually more reliable (rather than unreliable) than that at a regional or local scale.

Thank you for raising this point. We agree with your view. This sentence has been deleted in the revision.

Q13:L70: "Utilizing only one model will 'improve' the uncertainty of climate projection"?

Thank you for raising this point. We have modified this sentence as "Utilizing only one model will cause the rising in uncertainty of climate projection".

Q14: L90: "precious regional studies" -> "previous regional studies"

Thank you for raising this point. We have modified it in the revision.

Q15: Table 1: Please double-check the model info presented in Table 1. The grid spacing of MPIESM1-2-LR seems incorrect.

Thank you for raising this point. The resolution of MPIESM1-2-LR is 1.9°×1.9° which we have corrected

in the revision. We also have double-checked the model information (Table 1) in the revision.

Reference:

Makula E K, Zhou B. Coupled Model Intercomparison Project phase 6 evaluation and projection of East African precipitation. International Journal of Climatology, 42(4): 2398-2412, 2022.

Brovkin, Victo, Wieners, Karl-Hermann, Giorgetta, Marco, et al. Erich (2019). MPI-M MPIESM1.2-LR model output prepared for CMIP6 C4MIP. Version YYYYMMDD. Earth System Grid Federation, 2019. https://doi.org/10.22033/ESGF/CMIP6.748

Q16: L137: Why are there 540 observation images? How many years of observation data are used for input?

Thank you for raising this typo. It is a careless mistake which was happened in the process of writing paper and didn't reflect the conclusion. There are 360 observation images and we have corrected it in the revision. The period is 1965-2014 as Fig.1 shown. For each year, there are 12 observed images.

Q17: Fig.1: Which month is used, 01 or 02 (YYYY01 or YYYY02), in the second group of observation data?

Thank you for raising this typo. February is used in the second group. We have corrected it as "YYYY02" in the Fig.1.

[Figure]

Fig. 1. Weight assignment of 16 GCMs on a time scale

Q18: L143: "a widely technique applied for" -> "a technique widely applied for"

Thank you for raising this point. We have modified it in the revision.

Q19: L146: There are two "Lee et al., 2022" in the reference list. Please clarify which one you referred to here.

Thank you for raising this point. We have clarified it using "a" label which can be read as "which determine the relationship between one or more independent quantitative variables and another variable (Lee et al., 2022a)".

Reference:

Lee, J., Lee, W.S., Jung, H. and Lee, S.-G.: Comparison between total least squares and ordinary least squares in obtaining the linear relationship between stable water isotopes. Geoscience Letters, 9, 11, 2022a.

Q20: L172-173: "following equation 4"?

Thank you for raising this typo, and we have corrected the number of the equation in the revision.

Q21: L191: delete "neural network"

Thank you for raising this point. We have deleted it in the revision.

Q22: L210: delete "ranging from -1 to 1"

Thank you for raising this point. We have deleted it in the revision.

Q23: L211-214: On what basis were correlation coefficients divided into five categories? The range of correlation coefficients is also affected by the size of the samples. A significance test is more desirable than a specified range of correlation coefficients. Is it appropriate to define no correlation as R=0 strictly?

Thank you for raising this point. R is a measure between a random variable y and its prediction from a regression model. Correlation coefficients divided into five categories according to the fitting degree of samples ($x_i$, $y_i$) and least squares regression line. The $x_i$ and $y_i$ denoted the $i^{th}$ value of variable x and y, respectively. We agree with your comments. Considering the categories have not been applied in the accuracy assessment, we have deleted the categories and definition in the revision.

Reference:

Asuero, A.G., Sayago, A. and González, A.G.: The Correlation Coefficient: An Overview. Critical Reviews in Analytical Chemistry, 36, 41-59, 2006.

Q24: L226: Please double-check the citation of equations here.

Thank you for raising this typo. We have corrected the number of the equation in the revision.

Q25: L236-243, L329-331: The authors define the p$^{th}$ index with R, CRMSE, SD, and MAE. Note that these statistics are not independent of each other. For example, CRMSE is a function of R and SD (Taylor 2001; Xu et al., 2019). In another word, CRMSE already includes R and SD. Thus, it is not necessary to combine these statistics and compute CRI. However, the authors may consider computing CRI using the temperature and precipitation indices and compute CRI, which represents the model's overall ability to simulate both variables.

Reference:

Taylor, 2001: Summarizing multiple aspects of model performance in a single diagram. JGR-Atmos, 7183-7192

Xu et al, 2019: Comments on 'DISO: A rethink of Taylor diagram'. IJoC.40, 2506-2510

Thank you for your questions. We have carefully read the references. R, SD ratio and CRMSE are all calculated by observed and prediction dataset. Xu et al. (2019) proposed CRMSE is a function of R and SD, however, CRMSE is not a function of R and SD ratio. SD ratio calculated by SD of observation and prediction dataset was applied to assess the overall accuracy rather than SD.

Q26: 3: The figure appears of poor quality. It is extremely hard to distinguish different models! The authors should also clarify SD and cRMSE were normalized by observed SD. Are the authors showing climatological mean temperature and precipitation? Over which region? Which season? Only for the land area? Please clarify.

Thank you for raising these points. We have modified Fig.3 and improve its resolution into 600dpi. We have also clarified the point that SD and CRMSE were normalized by observed SD in the revision. These indices represent the accuracy of GCMs and ensemble model dataset (1995-2014) against observation (CRU TS4.5) over global continent corresponding to the year-to-year temporal variation, respectively. There are 67420 pixels in each image which represent the global continent. For example, the CRMSE value of BCC-CSM2-MR in Taylor Diagram was calculated as follows:

(1) According to the year-to-year and temporal relationship, get the pairs of value of BCC-CSM2-MR and observed dataset in January. The time period was January of 1994-2015. CRU TS4.5 dataset (1994-2015) was applied as the observed dataset. Therefore, there are 1348400 pairs (20 years* 1 image/year* 67420 pixels/image) of GCM and observed values.

(2) Repeat step1, get pairs of values in other 11 months. The sum of pairs is 16180800(1348400 pairs/month *12 months).

(3) Calculate CRMSE through 16180800 pairs to represent the R value of BCC-CSM2-MR in Taylor diagram.

Figure 3 is shown as follows.

[Figure]

Fig. 3. Taylor diagrams of (a) temperature and (b) precipitation. Ref stands for CRU TS4.05 observation dataset. SD and CRMSE were normalized by observed SD.

Q28: 4: Similar to Fig. 3, the figure caption failed to provide enough information. Does the figure show annual mean or seasonal mean precipitation (temperature)? Which region? Are the MAE and percentile calculated based on a climatological mean spatial field or a time series?

Thank you for raising these points. We have modified Fig.4 and improve its resolution into 600dpi. There are 67420 pixels in each image of observed and GCM dataset which represent the global continent. For

the $i^{th}$ spatial location ($1<i\leq67420$), we get the values of $i^{th}$ location from GCM dataset and observed dataset (1995-2014) in each month corresponding to temporal relationship. Therefore, there are 240 pairs for the $i^{th}$ location to calculate the MAE of this pixel. For each GCM, the spatial pattern of MAE can be drawn as a map according to 67420 MAEs of pixels (Fig. 5). Finally, we percentile of MAE of the $j^{th}$ GCM according to the 67420 values which represent the MAE of each spatial location.

Q29: L293-295, L207-209: why? Could you please explain the reason?

Thank you for raising these points.

Murphy, J. et al. (2004) also found that, with increase of latitude in the northern hemisphere, the upper regions of the northern hemisphere owned higher density of error. They proposed the spread in surface warming predictions is dominated by process uncertainties. However, for precipitation, internal variability makes a much larger contribution, particularly in extratropical regions.

Refer

Murphy, J., Sexton, D., Barnett, D. et al. Quantification of modelling uncertainties in a large ensemble of climate change simulations. Nature, 2004, 430, 768–772. https://doi.org/10.1038/nature02771

Q30: L343: "overall pattern" -> "overall spatial variability"

Thank you for raising this point. We have corrected it in the revision.

Q31: L345: This is not true.

Thank you for raising this issue. We have deleted this sentence.

Q32: 8: Please use thick lines for the global mean temperature anomalies.

Thank you for raising this point. We have used thick lines global mean temperature anomalies in Figure 8 which can be shown as follows.

[Figure]

Fig. 8. Temperature anomalies of global and continents under (a) SSP1-2.6 (b) SSP2-4.5 and (c) SSP5-8.5 respect to pre-industrial temperature (1850-1900). N. Am denotes North America. S. Am denotes South America.

Q33: L435-436: Please double check the citation of Fig. 10a and 10b.

Thank you for raising this typo.

Q34: L441: "What's more" -> "Moreover"

We have modified it in the revision.

Q35: L442: delete "with"

We have modified it in the revision.

Q36: L443-444: The sentence does not read well.

Thank you for raising this point. The sentence has been modified to "Similarly, relatively abundant precipitation in Europe and North America will also increase under SSP5-8.5".

Q37: L469: "decrease" -> "reduce"

Thank you for raising this point. We have modified it in the revision.

Q38: L493: "quicker" -> "slower"

Thank you for raising this typo.

Q39: L512: "aggravate" -> "generate a stronger"

We have modified it in the revision.

Q40: L522: "intervention" -> "mitigation"

We have modified it in the revision.

Q41: L535, 538: "expected to accurately project climate change", "high credible findings". Climate projection is strongly affected by scenario uncertainty, model uncertainty, and internal climate variability. Given these uncertainties, it is impossible to accurately project climate change using a multi-model ensemble mean. Thus, uncertainty range estimation is also of great importance.

Thank you for raising these points. We have corrected them in the revision. For example, "expected to accurately project climate change" has been modified to "expected to improve the accuracy of projecting climate change" and "high credible findings" has been modified to "findings". We strongly agree with your views, uncertainty range estimation was written as the caveats in the discussion.